# Revealing Behavioral Plasticity in Large Language Models:
# A Token-Conditional Perspective

**Liyuan Mao** [1 2]  **Le Yu** [1]  **Jing Zhou** [1]  **Chujie Zheng** [1]  **Bowen Yu** [1 †]  **Chang Gao** [1]  **Shixuan Liu** [1]  **An Yang** [1]
**Weinan Zhang** [2]  **JunYang Lin** [1]

## Abstract

In this work, we reveal that Large Language Models (LLMs) possess intrinsic behavioral plasticity—akin to chameleons adapting their coloration to environmental cues—that can be *exposed* through token-conditional generation and *stabilized* via reinforcement learning. Specifically, by conditioning generation on carefully selected token prefixes sampled from responses exhibiting desired behaviors, LLMs seamlessly adapt their behavioral modes at inference time (e.g., switching from step-by-step reasoning to direct answering) without retraining. Based on this insight, we propose **To**ken-**Co**nditioned **R**einforcement **L**earning (**ToCoRL**), a principled framework that leverages RL to internalize this chameleon-like plasticity, transforming transient inference-time adaptations into stable and learnable behavioral patterns. ToCoRL guides exploration with token-conditional generation and keep enhancing exploitation, enabling emergence of appropriate behaviors. Extensive experiments show that ToCoRL enables precise behavioral control without capability degradation. Notably, we show that large reasoning models, while performing strongly on complex mathematics, can be effectively adapted to excel at factual question answering, which was a capability previously hindered by their step-by-step reasoning patterns.

## 1. Introduction

Chameleons possess a remarkable ability to adapt their appearance in response to environmental stimuli: shifts in light, temperature, or surroundings can trigger dynamic changes

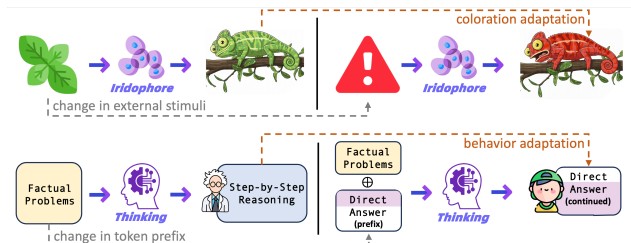

*Figure 1.* Like chameleons changing their coloration in response to external stimuli (top), language models can adapt their behavior according to the token prefix (bottom). ⊕ stands for concatenation.

in their coloration. This adaptive behavior reflects an intrinsic plasticity: the capacity exists in the organism, waiting to be expressed when appropriate cues are present. Similar to chameleons, Large Language Models (LLMs) also exhibit such plasticity (Brown et al., 2020; Ouyang et al., 2022), which serves as the cornerstone for building variant specialized models with diverse behaviors (Gururangan et al., 2020; OpenAI, 2025; Yang et al., 2025; Guo et al., 2025).

Typically, controllable behavioral steering of LLMs relies on parameter-update methods, such as supervised fine-tuning or preference optimization (Radford et al., 2018; Grattafiori et al., 2024; Rafailov et al., 2023). These approaches merely elicit or amplify pre-existing behavioral patterns already embedded in the training data (Chung et al., 2024; Yang et al., 2025), rather than generating genuinely novel behaviors. In stark contrast, we uncover a distinct form of behavioral plasticity in LLMs that requires *no* parameter updates at inference time yet enables the emergence of behaviors extending *beyond* those implicitly present in the training data. This plasticity can be systematically exposed through token-conditional generation and subsequently stabilized via reinforcement learning, revealing latent capacities not directly encoded in the original training distribution.

Specifically, by conditioning the generation process on carefully selected token prefixes—derived from responses exhibiting desired behaviors—the language model can adapt its behavior accordingly, without any parameter modification. Take Large Reasoning Models (LRMs) as a concrete example. These models excel at complex mathematical

[1]Qwen Team, Alibaba Group [2]Shanghai Jiao Tong University. Correspondence to: Bowen Yu <yubowen.ph@gmail.com>, Weinan Zhang <wnzhang@sjtu.edu.cn>.

*Proceedings of the 43rd International Conference on Machine Learning*, Seoul, South Korea. PMLR 306, 2026. Copyright 2026 by the author(s).

problem-solving through step-by-step reasoning (Guo et al., 2025; OpenAI, 2025). However, they often performs significantly worse than their instruct counterparts on straightforward factual questions, which directly answer the questions (Yang et al., 2025). By conditioning response generation on prefixes from such direct answers (as shown in Fig. 2), the same LRM shifts from step-by-step reasoning to direct knowledge retrieval. This uncovers a fundamental property of auto-regressive language models: behaviors emerge dynamically through the interaction between model parameters and the generated context. The initial tokens establish a behavioral trajectory that the model can naturally continue, suggesting that LLMs function as adaptive systems where outputs reflect both learned knowledge and contextual behavioral cues. When examined on a concrete model, such behavior adaptation instantly improves the accuracy of Qwen3-30B-A3B-2507-Thinking on SimpleQA benchmark (from 18.9 to 20.7), suggesting the potential to build a general-purpose language model that can adopt the most suitable behavior for different types of problems.

While token-conditional generation effectively adapts the behavior, relying on it at inference time requires access to appropriate prefix examples. Besides, the adaptive behavior is transient and unstable, depending on external conditioning signals. To transform this exposed plasticity into persistent capability, we propose **To**ken-**Co**nditional **R**einforcement **L**earning (**ToCoRL**), a framework that *stabilizes* the chameleon-like adaptability through reinforcement learning. ToCoRL internalizes token-conditional behavioral control in a principled manner, enabling models to autonomously execute appropriate behaviors without inference-time guidance. Unlike standard RLVR process (Shao et al., 2024; Williams, 1992), ToCoRL incorporates token-conditional generation in the roll-out stage. Apart from the policy-improvement objective, ToCoRL leverages these token-guided rollouts to compute a customized KL divergence that shapes exploration toward desired behaviors while still enhancing exploitation. Consequently, ToCoRL induces the emergence of the most suitable behavior and further stabilizes it during the RL process, transforming ephemeral adaptations into persistent capabilities that models learn to invoke autonomously.

To demonstrate the effectiveness of our approach, we focus on a challenging test case: adapting LRMs to excel at both complex mathematical reasoning and factual question answering—two tasks requiring fundamentally different behavioral strategies. Training with ToCoRL leads to the emergence of a novel factual-answering behavior, which shares the best of explicit reasoning and direct-answering behavior. Starting from Qwen3-30B-A3B-2507-Thinking, ToCoRL improves accuracy on the SimpleQA benchmark from 18.9 to 28.3. Moreover, the emergent behavior coexists with mathematical reasoning capabilities without interfer-

ence, improving accuracy on the AIME'25 benchmark from 80.5 to 81.5, thereby demonstrating that diverse behaviors can be stabilized within a unified model. Through extensive experiments and analysis, we provide strong evidence that ToCoRL is necessary for the emergence of such behavior and shed light on the underlying mechanisms. Moreover, when building a versatile LRM from a base model, the discovered behavior pattern can be easily transferred through supervised fine-tuning.

Our work establishes token-conditional generation as a fundamental lens for understanding behavioral adaptability in LLMs. Different specialized models are not entirely distinct entities; their performance differences do not necessarily stem from gaps in knowledge or capability, but may instead arise from differences in behavioral patterns and how models approach problems. This suggests a paradigm shift: instead of training separate specialized models, we can program diverse behaviors within unified models by learning to control the token-level patterns that guide behavioral expression. By revealing and harnessing this chameleon-like adaptability, we move closer to truly general-purpose AI systems that flexibly adapt their problem-solving strategies to match diverse task demands.

**Conflict of Interest Disclosure** The author Liyuan Mao conducted this work during an internship at Qwen Team, Alibaba Group, which develops the Qwen models used in this paper.

## 2. Preliminaries

**Notations in RL for LLMs.** LLMs have long been viewed as policies and optimized via RL (Ouyang et al., 2022; Shao et al., 2024; Yang et al., 2025), with $\pi$ commonly used to denote the LLM policy. There are two common definitions of actions: sequence-level action and token-level action. The former treats the entire response sequence as an action $a$, with the response space as the action space $\mathcal{A}$. The latter treats each token as an action, with the vocabulary as the action space. In this paper, we focus on the sequence-level action setting, where the state $s$ corresponds to the complete prompt and the state space $\mathcal{S}$ consists of diverse prompts. The reward $r(s, a)$ maps a prompt–response pair to a scalar value, and the value function $V^\pi(s)$ and advantage function $A^\pi(s, a)$ are defined as follows,

$$V^\pi(s) = \mathbb{E}_{a \sim \pi}[r(s, a)], \ A^\pi(s, a) = r(s, a) - V^\pi(s).$$

**REINFORCE Algorithm.** Being a classic on-policy RL algorithm, REINFORCE was first developed in traditional RL (Williams, 1992; Sutton et al., 1999) and later shown effective for LLMs (Ahmadian et al., 2024). In traditional RL settings, REINFORCE derives its objective by taking the

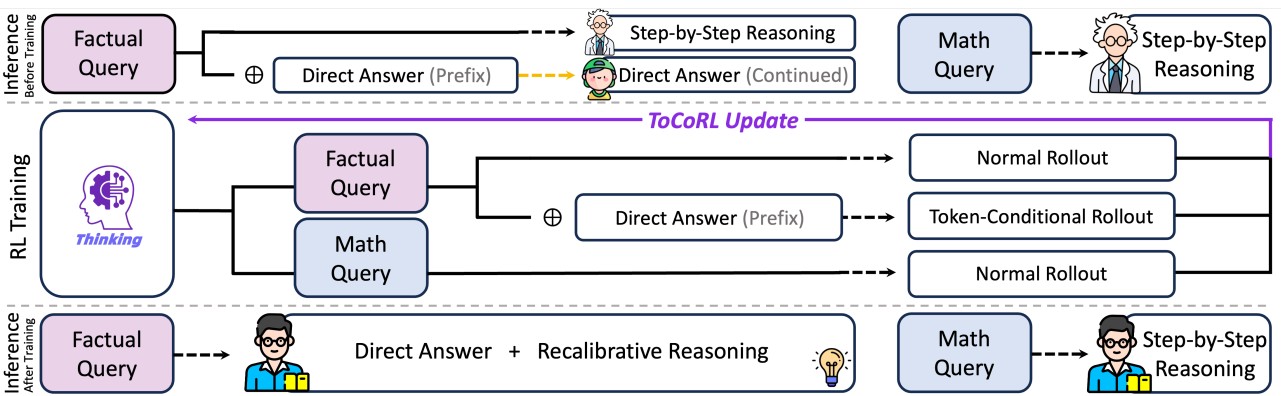

*Figure 2.* **Top**: Conditioning response generation on a direct-answer prefix enables LRM to switch from step-by-step reasoning to direct knowledge retrieval, revealing a new perspective on language model behavioral plasticity. **Middle**: ToCoRL leverages token-conditional generation to guide exploration during RL training, inducing the emergence of new factual answering behaviors. **Bottom**: After an initial direct answer, the ToCoRL-trained model adopts an emergent recalibrative reasoning behavior for factual problems (see Section 5.2), while its complex math problem–solving behavior remains unchanged. We provide concrete examples for further illustration in Fig. 4.

derivative of the expected return $\mathcal{J}(\pi_\theta)$ with respect to the parameters $\theta$, where $\mathcal{J}(\pi_\theta)$ is defined as follows. Here, $\tau$ denotes the rollout trajectory $(s_0, a_0), \ldots, (s_T, a_T)$, and $r_t$ is the reward at step $t$. The original REINFORCE objective, denoted $J(\theta)$, is also given below. We note that throughout this paper, a policy without a subscript denotes a detached version of its subscripted counterpart (e.g., $\pi$ is a detached $\pi_\theta$). In the sequence-level action setting, the trajectory length is 1 for single-turn dialogues, so only the first step needs to be considered in this objective,

$$\mathcal{J}(\pi_\theta) = \mathbb{E}_{\tau \sim \pi(\cdot|s)}\Big[\sum_{t=0}^{T} \gamma^t \cdot r_t\Big],$$

$$J(\theta) = \mathbb{E}_{\tau \sim \pi(\cdot|s)}\Big[\sum_{t=1}^{T} A^\pi(s, a) \log \pi_\theta(a|s)\Big].$$

## 3. Exposing Behavioral Plasticity via Token-Conditional Generation

To better demonstrate the behavioral plasticity of LLMs, we choose a language model whose behavior is already highly specialized. This shows that even when a model's behavior has been strongly specialized, its behavioral plasticity is still preserved and can be exposed with appropriate methods. Therefore, we adopt Qwen3-30B-A3B-2507-Thinking, a widely used LRM designed for complex reasoning tasks. It conducts step-by-step reasoning (wrapped by <think> and </think>), which is exploratory in nature and often involves trial-and-error or associative content, before producing the final answer. As shown in Table 7, such reasoning process boosts its capability to solve complex math problems by decomposing them into a sequence of simpler subproblems. On the other hand, this LRM performs much worse on factual questions compared with its instruct counter-

part, Qwen3-30B-A3B-2507-Instruct. Unlike the Thinking model, the Instruct model directly answers the questions without additional reasoning.

We introduce **token-conditional generation** to steer the behavior of the Thinking model. Specifically, given a factual problem, we first extract the initial tokens of the direct answer generated by the Instruct model. Conditioning reasoning context on this prefix, we then apply the Thinking model to continue generating subsequent tokens. To avoid answer leakage, the number of enforced tokens is sufficiently small while still forming a direct attempt to answer (e.g., 3 tokens). We evaluate the effect of token-conditional generation with SimpleQA (Wei et al., 2024), a commonly used factual problem benchmark. The results are reported in Table 1. Concrete examples, along with a detailed demonstration of the procedure, are provided in Appendix C.

*Table 1.* The following table shows the accuracy (avg@8), the average response length (Avg. length) and medium response length (Med. length) in token number, when tested on SimpleQA benchmark. Note that we use 'Normal Gen' and 'ToCo Gen' to denote normal generation and token-conditional generation respectively.

|                | Accuracy | Avg. length | Med. length |
|----------------|----------|-------------|-------------|
| The Normal Gen | 18.9%    | 1255        | 1128        |
| ToCo Gen       | 20.7%    | 891         | 477         |

The examples clearly demonstrate that, through token-conditional generation, the Thinking model tends to answer factual questions directly, skipping step-by-step reasoning. Table 1 shows that once its behavior is shifted, the Thinking model's performance on factual questions also improves, with significant decrease in the response length. This inference-time editing requires no additional training and introduces no new knowledge, so the performance gains and the decrease of response length can only be attributed

to the model's inherent behavioral plasticity.

Specifically in this case, the Thinking model's original reasoning style is not helpful for factual queries and can even hinder knowledge retrieval. Unlike complex math problems that require step-by-step reasoning to bridge the abstract problem and the correct answer, factual questions only require instruction-following skills, and the question-answer mapping usually exists in the pre-training corpus. Without controlling the reasoning process, unnecessary associations and unverified information, including potential hallucinations, may be introduced and mislead the model. Token-conditional generation leverages the behavioral plasticity, adapting the model's behavior to answer directly and consicely, thereby improving performance.

# 4. ToCoRL: Internalize Behavior Adaptation with Guided Exploration

The aforementioned behavioral plasticity endows a single language model with the potential to exhibit diverse behaviors. Nevertheless, this inference-time operation relies on pre-generated responses, and its effects are unstable and transient; moreover, it does not provide sufficient control over the subsequent tokens after token-forcing. To remedy this, we propose a novel RL-based method, Token-Conditional Reinforcement Learning (ToCoRL), which internalizes the above behavior adaptation, transforming transient behavior shifts into stable, learnable behavioral patterns.

## 4.1. Incorporate Guidance from Token-Conditional Generation via KL-Divergence

We develop ToCoRL from a KL-constrained RL objective, incorporating guidance from token-conditional generation to discover the most suitable behavioral pattern. KL divergence with a specific policy has long been used to guide exploration in on-policy RL (Rajeswaran et al., 2017; Nair et al., 2020). For example, when taking expert policy as the reference, the augmented KL divergence effectively guides the policy toward high-reward regions of the strategy space (Peng et al., 2018). In our setting, although the policy induced by token-conditional generation exhibits desirable behaviors, it does not qualify as an expert policy. As a result, we formulate our optimization objective below, where the KL divergence is calculated with a customized policy.

$$
\max_{s \sim \mathcal{D}} \mathbb{E}\Big[ \mathbb{E}_{a \sim \pi}[A^\pi(s,a) \log \pi_\theta(a|s)] - \lambda \mathrm{KL}(\tilde{\pi}_{\mathrm{TC}} || \pi_\theta) \Big],
$$

$$
\forall s, a \;\; \tilde{\pi}_{\mathrm{TC}}(a|s) = \pi(a|s) + \rho_s \cdot \pi_{\mathrm{TC}}(a|s) \cdot A^{\pi_{\mathrm{TC}}}(s,a), \quad (1)
$$

$$
\forall s \;\; \rho_s = \min_{a \in \mathcal{A},\, A^{\pi_{\mathrm{TC}}}(s,a) < 0} \Big\{ \frac{-\pi(a|s)}{\pi_{\mathrm{TC}}(a|s) \cdot A^{\pi_{\mathrm{TC}}}(s,a)} \Big\}.
$$

The objective contains two terms, the first being the REIN-FORCE objective commonly used in RL for LLMs. (Ahmadian et al., 2024; Schulman et al., 2017). In the second KL divergence term, the reference policy $\tilde{\pi}_{\mathrm{TC}}$ is the composition of the current policy $\pi$ and the policy induced from token-conditional generation, i.e. $\pi_{\mathrm{TC}}$. Despite its seemingly complex form, $\tilde{\pi}_{\mathrm{TC}}$ is well-defined under mild conditions, as established by the following theorem.

**Theorem 4.1.** *Given that for every state $s$, there exists an action $a$ such that $A^{\pi_{\mathrm{TC}}}(s,a) \neq 0$, the policy $\tilde{\pi}_{\mathrm{TC}}(a|s)$ is well-defined.*

We provide the proof in Appendix F. Since $A^{\pi_{\mathrm{TC}}}(s,a) = 0$ holds for all $a$ only when all answers yield the same reward, this assumption holds in almost all cases as long as the query is meaningful. We then proceed to justify the advantage of the optimization objective along two main aspects.

First, the interaction between the policy gradient term and the KL constraint term drives flexible and stable behavior adaptation. Early in training, the KL term injects the target behavior, inducing behavioral imitation of correct responses from token-conditional generation. As the model's behavior gradually shifts, standard rollouts in the policy gradient term also exhibit the target behavior. As a result, the policy gradient term internalizes the externally acquired behavior into an intrinsic model behavior that facilitates policy improvement, which in turn improves token-conditional generation. The KL coefficient regulates off-policy gradients to ensure training stability and avoid unintended behavior shifts.

Second, the KL divergence enables correctness-aware guided exploration in a computationally tractable manner. From the definition of $\tilde{\pi}_{\mathrm{TC}}$ in Eq. (1), it's evident that $\tilde{\pi}_{\mathrm{TC}}(a|s) > \pi(a|s)$ when $A^{\pi_{\mathrm{TC}}}(s,a) > 0$, and $\tilde{\pi}_{\mathrm{TC}}(a|s) < \pi(a|s)$ when $A^{\pi_{\mathrm{TC}}}(s,a) < 0$. In the 0/1 reward setting, a response has a positive advantage if and only if it is correct (Shao et al., 2024). Consequently, responses explored by $\tilde{\pi}_{\mathrm{TC}}$ are reinforced if they yield right answers and suppressed otherwise, thereby biasing exploration toward correctness.

## 4.2. Practical Implementation

In practice, while direct sampling from $\tilde{\pi}_{\mathrm{TC}}$ to compute the KL divergence is intractable, the following theorem provides an alternative implementation that relies only on samples from $\pi_{\mathrm{TC}}$, making the objective tractable.

**Theorem 4.2.** *From the perspective of the energy function underlying $\pi(\cdot|s)$, the following two objectives share the same gradient.*

$$
\begin{aligned}
&\text{Obj. 1:} \quad -\mathrm{KL}(\tilde{\pi}_{\mathrm{TC}}(\cdot|s) || \pi_\theta(\cdot|s)), \\
&\text{Obj. 2:} \quad \rho_s \cdot \mathbb{E}_{a \sim \pi_{\mathrm{TC}}(\cdot|s)}[A^{\pi_{\mathrm{TC}}}(s,a) \cdot \log \pi_\theta(a|s)].
\end{aligned} \quad (2)
$$

The proof is given in Appendix F. Theorem 2 quantitatively characterizes the correctness-aware property of the

customized KL divergence, using a tractable policy-gradient-style objective. We note that when $\pi(a|s)$ is an LLM with billions of parameters, analyzing exact gradients for all parameters becomes infeasible. Nevertheless, the analysis from the perspective of the energy function is generally applicable to any discrete distribution. Based on Theorem 2, we rewrite Eq. (1) into the following objective,

$$
\begin{aligned}
\max \mathbb{E}_{s \sim \mathcal{D}} \big[ & \mathbb{E}_{a \sim \pi(\cdot|s)}[A^{\pi}(s,a) \cdot \log \pi_{\theta}(a|s)] \\
& + \lambda \cdot \rho_s \cdot \mathbb{E}_{a \sim \pi_{\mathrm{TC}}(\cdot|s)}[A^{\pi_{\mathrm{TC}}}(s,a) \cdot \log \pi_{\theta}(a|s)] \big].
\end{aligned} \tag{3}
$$

To avoid separately estimating two advantages (i.e., $A^{\pi}(s,a)$ and $A^{\pi_{\mathrm{TC}}}(s,a)$), we merge the two policy-gradient objectives in Eq. (3) into a single surrogate objective in Eq. (4), with $\pi_{\mathrm{mix}}$ defined below. Response generation and advantage estimation in Eq. (4) relies solely on a mixed policy. This significantly reduces the variance of advantage estimation, especially when the advantages are estimated via Monte Carlo sampling (as in our implementation), and the number of samples is usually limited due to computational constraints. To justify its soundness, we first note that the objectives in Eq. (3) and Eq. (4) share the same rollout distribution. Moreover, Theorem 4.4 shows that the gap between the objective in Eq. (3) and its surrogate in Eq. (4) is well controlled. To this end, we define $\pi_{\mathrm{mix}}$ and $\phi_s$ below.

**Definition 4.3.** We define the mixed policy $\pi_{\mathrm{mix}} := \frac{\pi + \tilde{\lambda}_s \cdot \pi_{\mathrm{TC}}}{1 + \tilde{\lambda}_s}$, and the scalar constant $\phi_s := \min\{\frac{\tilde{\lambda}_s}{(1+\tilde{\lambda}_s)^2}, \frac{1}{(1+\tilde{\lambda}_s)^2}\}$. Note that $\tilde{\lambda}_s = \lambda \cdot \rho_s$.

**Theorem 4.4.** $\forall s \in \mathcal{D}$, define $\mathcal{A}_s^+ \subseteq \mathcal{A}$ as the set of correct responses for query $s$. The following inequality holds:

$$
\begin{aligned}
& \mathbb{E}_{a \sim \pi_{\mathrm{mix}}} \big[ \mathbb{1}[a \in \mathcal{A}_s^+] \cdot A^{\pi_{\mathrm{mix}}}(s,a) \cdot \log \pi_{\theta}(a|s) \big] \leq \phi_s \cdot \\
& \Big[ \mathbb{E}_{a \sim \pi(\cdot|s)} \big[ \mathbb{1}[a \in \mathcal{A}_s^+] \cdot A^{\pi}(s,a) \cdot \log \pi_{\theta}(a|s) \big] + \\
& \quad \tilde{\lambda}_s \cdot \mathbb{E}_{a \sim \pi_{\mathrm{TC}}(\cdot|s)} \big[ \mathbb{1}[a \in \mathcal{A}_s^+] \cdot A^{\pi_{\mathrm{TC}}}(s,a) \cdot \log \pi_{\theta}(a|s) \big] \Big].
\end{aligned}
$$

The proof is provided in Appendix F. Theorem 4.4 implies that, for all correct answers, the surrogate objective in Eq. (4) can lower-bounds the objective in Eq. (3), thereby providing a guarantee for its maximization. Intuitively, this connection arises because $A^{\pi}$, $A^{\pi_{\mathrm{TC}}}$, and $A^{\pi_{\mathrm{mix}}}$ share the same signs under 0/1 rewards—positive for correct responses and negative otherwise. In conclusion, we present the practical optimization objective below, where $\pi_{\mathrm{mix}}$ remains as previously defined. For simplicity of implementation, $\tilde{\lambda}_s$ is set to a fixed positive value $\lambda$ across all states.

$$
\begin{aligned}
& \max \mathbb{E}_{s \sim \mathcal{D}} \big[ \mathbb{E}_{a \sim \pi_{\mathrm{mix}}}[A^{\pi_{\mathrm{mix}}}(s,a) \cdot \log \pi_{\theta}(a|s)] \big], \\
& A^{\pi_{\mathrm{mix}}}(s,a^i) = r(s,a^i) - \mathrm{mean}\big(\{r(s,a^i)\}_{i=1}^{G}\big).
\end{aligned} \tag{4}
$$

## 4.3. Algorithmic Instantiation: Evolving Reasoning Behavior for Factual Problem Solving

We leverage ToCoRL to build a LRM that achieves superhuman performance factual problem solving, a setting where prior reasoning models often underperform. ToCoRL internalizes the performance-enhancing behavior shift described in Section 3 and induces the emergence of a novel reasoning behavior suitable for factual problems. Specifically, $\pi$ is set as a LRM and another instruct model $\pi_{\mathrm{Inst}}$ provides the token prefixes. The token-conditional generation process is illustrated in Appendix C and formulated as follows,

$$
\begin{aligned}
& \tilde{o}_{1\ldots k} := \texttt{<think>}\backslash\mathrm{n} \oplus o'_{1\ldots k-2}, \ o'_{1\ldots k-2} \sim \pi_{\mathrm{Inst}}(O'_{1\ldots k-2}|s), \\
& o \sim \pi_{\mathrm{TC}}(O|s) := \tilde{o}_{1\ldots k} \oplus o_{>k}, \ o_{>k} \sim \pi(O_{>k}|s, \tilde{o}_{1\ldots k}).
\end{aligned}
$$

To avoid confusion with advantage, we use $O$ to denote the response as a random variable. $\oplus$ denotes concatenation. We set the <think> token as the first token of the prefix, which should be discarded when applying ToCoRL to other model types. Since rule-based rewards are not applicable for factual problems, we employ LLM-as-a-judge to provide reward signals. The reward is binary (0 or 1), taking value 1 only when the generated answer is equivalent to the ground truth. The evaluation details and the pseudo-code of this instantiation are provided in Section E.2.

Note that ToCoRL is not limited to enhancing factual problem solving capability. For other tasks where the desired behavior can be encoded in token prefixes, ToCoRL can take effect by seamlessly injecting the target behavior and evolving the most effective behavioral pattern.

## 5. Experiments

We focus the experiments on the aforementioned test case: adapting an LRM's behavior to perform well on both complex reasoning tasks and factual problem solving. To demonstrate the effectiveness and underlying mechanisms of ToCoRL, as well as its implications for building versatile reasoning models, we address several key research questions. These questions can be grouped into four aspects as below.

**RQ1:** What performance improvements and behavioral changes does ToCoRL bring to LRMs?

**RQ2:** Through what underlying mechanisms do the performance improvements and behavioral changes arise?

**RQ3:** Are performance improvements and behavioral adaptations robust to changes in algorithmic factors? (e.g., hyperparameters, alternative prefix providers)

**RQ4:** Are the emergent behaviors transferable and exploitable by other models to solve factual problems?

## 5.1. Effective Performance Improvement and Emergent Reasoning Behavior (RQ1)

We use Qwen3-30B-A3B-2507-Thinking as the starting point for training, and Qwen3-235B-A22B-2507-Instruct as the token prefix provider for token-conditional generation. Our training data consist of two components.

For factual problem solving, we use 10K factual query-answer pairs. In addition, to further strengthen complex mathematical reasoning capability and to examine the interaction between factual and math training, we jointly train on 10K math query-answer pairs. The evaluation is based on widely-used benchmarks comprising two corresponding components. For factual problems, we use SimpleQA benchmark (Wei et al., 2024) and AA-Omniscience benchmark (Jackson et al., 2025). For mathematical reasoning, we evaluate on AIME'24 and AIME'25. To mitigate evaluation variance given the limited data, we test average accuracy over 32 evaluations (Avg@32) for math queries.

To thoroughly demonstrate ToCoRL's effectiveness, we conduct a comparative study against various baseline RL-for-LLM algorithms. The baseline algorithms are as follows:

- GRPO: an representative RL algorithm, especially under the verifiable reward setting.
- Adaptive-Thinking: an RL algorithm biasing the response toward conciseness via a length-based advantage penalty.
- Instruct-Tuning GRPO: built on GRPO, we modify the rollout stage by adding a manually designed instruction at the end of each prompt, encouraging the LRM to reason adaptively according to the difficulty of the query.

We provide the details of the baseline algorithms, selection rationales, and additional experimental settings, in Appendix E. During the joint training process described above, we apply ToCoRL and the baseline methods only to factual-problem training, while using GRPO uniformly for training on math problems. In Table 2, we report the accuracy across different benchmarks after training process converge.

*Table 2.* Evaluation results for experiments based on Qwen3-30B-A3B-2507-Thinking. We use 'Thinking' and 'Instruct' to denote the corresponding two open-source models, and 'w.' to denote the training method used for factual problem queries. 'Adapt-Think', 'I-T GRPO', and 'Omnisci' are abbreviations for Adaptive-Thinking, Instruct-Tuning GRPO, and AA-Omniscience, respectively, and we consistently use these notations in subsequent tables.

| Qwen3-30A3B | SimpleQA | Omnisci | AIME'24 | AIME'25 |
|---|---|---|---|---|
| Thinking | 18.9 | 16.1 | 84.8 | 80.5 |
| Instruct | 22.1 | 10.5 | 75.2 | 61.3 |
| w. ToCoRL | **28.3** | **18.6** | 89.0 | **81.5** |
| w. GRPO | 23.6 | 16.7 | **89.5** | 81.4 |
| w. Adapt-Think | 23.9 | 16.0 | 86.9 | 78.9 |
| w. I-T GRPO | 21.1 | 14.8 | 87.8 | 80.4 |

As is shown in Table 2, ToCoRL demonstrates significant superiority in improving the capability of factual problem solving, compared with other baseline methods. While ToCoRL only marginally affects reasoning performance, Adaptive-Thinking and Instruct-Tuning GRPO incur noticeable performance drops on complex math benchmarks. Besides such quantitative results, we qualitatively analyze the change in behavioral patterns by examining different models' responses to the same factual problem set. Due to limited space, we present detailed analysis in Appendix G.

After training with ToCoRL, the LRM consistently starts its response with a direct answer. It then tends to perform recalibrative reasoning, where it repeatedly generates answers it currently deems to be correct, uses them as context for subsequent generation, and terminates with a summary once it becomes confident in the answer. Unlike Qwen3-30B-A3B-2507-Thinking, the ToCoRL-induced reasoning remains tightly focused on the question itself—not for proving or justification, but for retrieving the correct answer—thus avoiding unnecessary associations and unverified information. For complex math problems, ToCoRL induces only minor behavioral changes, consistent with its steady improvements on the AIME benchmarks. Intuitively, this is because token-conditional generation only produce content that is highly specific to the corresponding factual problems, which is hard to affect training on math queries.

## 5.2. Underlying Mechanisms of Performance and Behavioral Emergence in ToCoRL (RQ2)

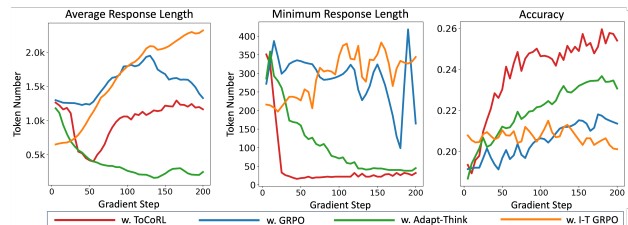

*Figure 3.* For clearer visualization, we plot the curves of these metrics over the first training epoch. Compared to ToCoRL, GRPO extends reasoning length while keeping the original reasoning style. Since unnecessary associations and unverified information persist, longer reasoning yields minor improvements, and response lengths eventually return to their initial levels. Adaptive-Thinking penalizes all reasoning, so evaluation scores rise quickly at first as ineffective context is removed. However, without additional reasoning capacity, the LRM's potential is underutilized, limiting further gains. Instruct-tuning RL initially shortens reasoning, but the prompt instruction gradually loses control over the behavior.

While ToCoRL yields substantial performance improvements and seemingly out-of-nowhere emergent behaviors, these outcomes are in fact driven by principled mechanisms. To analyze the mechanisms, a case-by-case analysis is infeasible due to the thousands of factual-problem queries used for training and evaluation. Nevertheless, the trends of

several response-length–based metrics, along with response examples from different stages of training, clearly explain these mechanisms. As a result, based on the experiments in Section 5.1, we trace these metrics and accuracy under SimpleQA benchmark along the training in Fig. 3. The response examples in different stages are presented in Appendix H.

We focus on two length-based metrics: the average and minimum response lengths across the evaluation dataset. Early in training (steps 0 to around 40), ToCoRL rapidly reduces both metrics. This occurs because token-conditional generation forces the LRM to produce a direct answer at the beginning, after which it tends to summarize, resulting in significantly shorter responses. As shown in Section 3, such behavior outperforms the original step-by-step reasoning on factual problems. The reward-maximizing nature of RL quickly reinforces the direct-answer behavior, further reducing both length metrics (supported by response examples in Appendix H). In other words, score improvements at this stage primarily stem from eliminating spurious reasoning content, adopting a concise reasoning behavior.

Subsequently, while the minimum response length stays low, the mean response length gradually increases. For simple problems, the LRM often retrieves the correct answer on the first attempt, and additional reasoning only hinders retrieval, keeping responses short. For difficult problems, one-shot retrieval fails, so the LRM extends the reasoning context to find the correct answer. RL continuously incentivizes the LRM to adopt the most effective reasoning behavior, leading it to revise answers using only the necessary context, thus preserving the advantage of the direct-answer behavior mentioned earlier. At this stage, ToCoRL guides the LRM in deciding when to trust its current answer candidate and when to generate alternatives, resulting in improved evaluation scores as reasoning capability is better utilized.

*Table 3.* Results for experiments based on Qwen3-30B-A3B-SFT, obtained by enabling the base model (denoted as 'Base') with a unified reasoning behavior. We refer to it as 'SFT' for brevity.

| Qwen3-30A3B | SimpleQA | Omnisci | AIME'24 | AIME'25 |
|---|---|---|---|---|
| Base | 24.2 | 12.0 | 15.1 | 9.4 |
| SFT | 19.6 | 14.0 | 56.5 | 38.7 |
| w. ToCoRL | **28.1** | **15.8** | **72.1** | 61.7 |
| w. GRPO | 22.6 | 14.3 | 72.0 | **61.9** |
| w. Adapt-Think | 24.8 | 13.8 | 47.9 | 30.4 |
| w. I-T GRPO | 21.7 | 12.9 | 56.0 | 37.8 |

One concern is that the aforementioned mechanisms may only apply to Qwen3-30B-A3B-2507-Thinking, or that the emergent behavior pre-exists in this model due to its diverse training data. To address this, we start from the base model and use SFT to enable a unified reasoning behavior distinct from that learned by ToCoRL (details in Appendix I). Such enabled behavior is associative, aiding mathematical rea-

soning but can hinder knowledge retrieval. We then repeat the experiments in Section 5.1. Results in Tab. 3 show that ToCoRL still substantially improves factual problem-solving performance, surpassing all baselines. As shown in Appendix J, the emergent behavior mirrors that in Section 5.1. Moreover, while Adaptive-Thinking reduces response length, unnecessary associations are still included in the limited reasoning context, highlighting the necessity of ToCoRL for achieving the desired reasoning behavior.

### 5.3. Robustness of ToCoRL (RQ3)

To validate the robustness of ToCoRL, we conduct an ablation study to examine the influence of key factors on the algorithm (e.g., hyperparameters, token prefix provider). As shown in the pseudo-code in Algorithm 1, the main tunable factors are KL coefficient $\lambda$, token prefix length $k$, and token prefix provider $\pi_{\text{Inst}}$. We test each factor individually while keeping all others consistent with the settings in Section 5.1. The evaluation scores are summarized below.

*Table 4.* Results for ablation studies on hyperparameters ($\lambda$ and $k$). We skip testing $k < 6$ since the required <think>\n prefix (occupying 2 tokens) makes the prefix too short to form a direct attempt at answering the problem.

| $(\lambda,\ k)$ | SimpleQA | Omnisci | AIME'24 | AIME'25 |
|---|---|---|---|---|
| $(1,\ 6)$ | 28.3 | 18.6 | 89.0 | 81.5 |
| $(1,\ 7)$ | 27.5 | 18.1 | 88.2 | 80.6 |
| $(1,\ 8)$ | 27.6 | 18.3 | 88.5 | 80.9 |
| $(0.5,\ 6)$ | 28.1 | 18.3 | 88.3 | 80.5 |
| $(2, 6)$ | 27.2 | 17.8 | 88.0 | 80.0 |

*Table 5.* Results for ablation studies on different prefix providers ($\pi_{\text{Inst}}$). For brevity, the 'Instruct' suffix is omitted for all models. Note that Qwen3 models in the table refer to the 2507 version.

| $\pi_{\text{Inst}}$ | SimpleQA | Omnisci | AIME'24 | AIME'25 |
|---|---|---|---|---|
| Qwen3-235A22B | 28.3 | 18.6 | 89.0 | 81.5 |
| Qwen3-30A3B | 28.0 | 18.6 | 89.1 | 81.3 |
| Qwen2.5-14B | 27.8 | 18.3 | 89.0 | 81.1 |

Table 4 clearly demonstrates ToCoRL's robustness to hyperparameters, with only minor performance fluctuations as the values of $\lambda$ and $k$ vary. The results in Table 5 show that, when using less capable models as token prefix providers, ToCoRL still achieves consistent improvements. Moreover, the emergent behavior is nearly identical across different choices of $\pi_{\text{Inst}}$. Intuitively, ToCoRL's efficacy is almost independent of $\pi_{\text{Inst}}$'s capacity, as long as the initial tokens attempt to provide a direct answer. Besides, while the prefix steers the initial direct-answer segment, the rest of the reasoning is fully determined by the model itself.

## 5.4. Transferring Emergent Behavior for Effective LRM Training (RQ4)

Previous experiments have demonstrated that, given an LRM, ToCoRL can boost factual problem-solving capability by adapting its reasoning behavior. However, it would be restrictive if every LRM had to undergo ToCoRL training to acquire the aforementioned behavior. A more common and effective strategy for acquiring specific behaviors is to first distill them into a dataset and then apply supervised fine-tuning from a base model. We show that the emergent behavior induced by ToCoRL is well suited to this strategy.

Specifically, we first construct a new SFT dataset based on the one used in the experiments in Table 3. The math-problem SFT data is kept unchanged, while the responses for factual problems are regenerated using the model trained via ToCoRL in Table 2. For comparison, we construct two additional datasets using Qwen3-235B-A22B-Instruct and Qwen3-30B-A3B-Instruct. These datasets share the same math SFT data, and for factual responses, the instruct models' outputs are treated as reasoning context, while summaries are generated by prompting another LLM to summarize the reasoning (details in Appendix K). Using these SFT datasets and starting from the same base model as in Table 3, we conduct SFT and run the same set of experiments. For a fair comparison and to minimize the influence of the RL algorithm on behavioral patterns, we consistently use GRPO for training. We report the scores in Table 6, presenting the training curves and response examples in Appendix K.

*Table 6.* We use 'Response Provider' to denote the source of the factual responses used for new SFT datasets as described above. All providers are from the Qwen3 series; [*] indicates manually constructed responses, otherwise are generated normally.

| Response Provider | SimpleQA | Omnisci | AIME'24 | AIME'25 |
|---|---|---|---|---|
| 30A3B-w. ToCoRL | **29.1** | **16.0** | **74.5** | **62.0** |
| 235A22B-Instruct[*] | 28.3 | 15.5 | 68.6 | 55.2 |
| 30A3B-Instruct[*] | 25.3 | 14.5 | 69.3 | 59.2 |

These results highlight the superiority of ToCoRL-discovered reasoning behavior in two aspects. First, the model's factual problem-solving capability improves significantly and immediately after SFT, without additional RL training. This highlights the strong transferability of the behavior and underscores its value for factual problem solving: although the regenerated responses are not guaranteed to be correct, the reasoning behavior itself enhances the base model's accuracy in knowledge retrieval. Second, the reasoning behavior is highly extensible. After the RL stage, the LRM equipped with the ToCoRL-discovered behavior achieves the largest accuracy gains and surpasses all baseline models. Moreover, it coexists compatibly with the behaviors in math-problem SFT data, leading to the greatest performance improvements on complex math benchmarks.

## 6. Related Works

**LLM Behavior Plasticity and Steering Methods** Pretrained on vast data sources, LLMs possess the plasticity to exhibit diverse behaviors depending on the elicitation method (Raffel et al., 2020; Yang et al., 2025). This plasticity stems from training billions of parameters on massive data, using text as a unified medium for learning and memorization (Chung et al., 2024). To steer model behavior, post-training techniques generally fall into two categories. The first involves parameter updates—such as SFT (Radford et al., 2018), RLHF (Ouyang et al., 2022), and Preference Optimization (Rafailov et al., 2023)—which controllably guide the model toward target behaviors pre-existing in the data. Conversely, prompt engineering and in-context learning steer behavior without parameter updates (Brown et al., 2020; Wei et al., 2022), offering flexibility while sacrificing controllability and stability. ToCoRL shares the advantages of both: it does not require the target behavior to pre-exist and provides precise control over the induced behavior.

**Reinforcement Learning Algorithms and Applications in LLM** Reinforcement learning algorithms typically optimizes an agent's objective of maximizing cumulative rewards (Schulman et al., 2015; Haarnoja et al., 2018). In the classic online setting, the agent interacts with an environment to collect data for policy updates (Mnih et al., 2015). This paradigm is well-suited for LLMs, where environment dynamics consist of simple token concatenation and rollouts are highly parallelizable (Ouyang et al., 2022; Qwen et al., 2025). Consequently, Proximal Policy Optimization (PPO) (Schulman et al., 2017) was among the first online RL algorithms applied to instruction tuning (Ouyang et al., 2022). Other methods, such as REINFORCE (Williams, 1992), have also proven effective for fine-tuning (Ahmadian et al., 2024). More recently, Group Relative Policy Optimization (GRPO) (Shao et al., 2024) and its variants (Yang et al., 2025; Guo et al., 2025) dispense with the traditional value network for advantage estimation, relying instead on relative rewards. This approach has demonstrated particular efficacy in domains with verifiable rewards.

## 7. Conclusion & Discussion

In this paper, we reveal the behavior plasticity of LLMs from a novel perspective. By leveraging token-conditional generation, a language model's behavior can seamlessly adapt to a different one, provided with a prefix of the response exhibiting the desired behavior. Based on this, we developed ToCoRL, a RL-based algorithm that stabilizes and internalizes such behavior adaptation, enabling the emergence of more effective behavioral patterns. Focusing on adapting the LRM's behavior to excel at both complex reasoning tasks and factual problem solving, we conduct extensive experiments to demonstrate ToCoRL's effectiveness, under-

lying mechanisms, robustness and transferability. Due to computational constraints, we do not evaluate ToCoRL on a broader range of tasks. Nevertheless, token-conditional generation is evidently applicable to a wide variety of settings. Our future work will extend ToCoRL to a wider range of tasks and different types of language models.

## Acknowledgement

The SJTU team was supported by National Natural Science Foundation of China (62322603).

## Impact Statement

This paper explores LLM behavioral plasticity from a novel perspective and introduces a behavior-steering method. Specifically for LRMs, we enhance their factual problem-solving capabilities, enabling them to excel in both complex mathematic problems and factual retrieval. Alongside these technical contributions, our approach may improves model reliability in other applications requiring high factual accuracy, such as education and decision support.

However, enhanced controllability may introduce risks if misused to steer models toward biased or harmful outputs. Specifically, behavior-steering mechanisms could be exploited to induce overconfident or selectively framed responses. These risks underscore the necessity of transparency and responsible deployment when applying these methods in real-world settings.

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

## A. Performance Difference Between Thinking and Instruct Models (Qwen3 Open-sourced Series)

*Table 7.* Comparison of accuracy on complex math benchmark (AIME'25) and factual problem benchmark (SimpleQA). In the same model series, the Thinking version, which conduct step-by-step reasoning before arriving at final answer, consistently perform better on complex math benchmark and worse on factual problem benchmark.

|  | Qwen3-235B-A22B-2507 | | Qwen3-30B-A3B-2507 | |
|---|---|---|---|---|
|  | Thinking | Instruct | Thinking | Instruct |
| AIME'25 | **92.3** | 70.3 | **80.5** | 61.3 |
| SimpleQA | 47.0 | **54.3** | 19.2 | **22.1** |

## B. Detailed Demonstration with Concrete Examples for Fig. 2

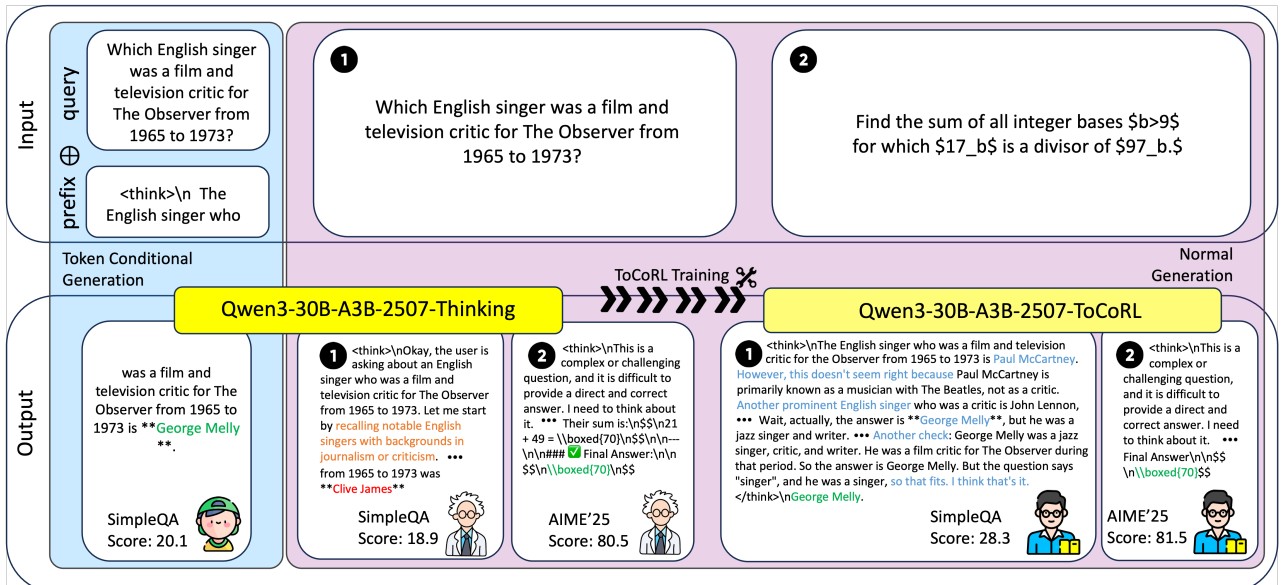

*Figure 4.* Detailed demonstration of the behavior change brought by token-conditional generation and ToCoRL. Concrete queries and responses are provided.

## C. Illustrative Diagram and Concrete Examples of Token-Conditional Generation

The illustrative diagram of token-conditional generation procedure is presented as follows.

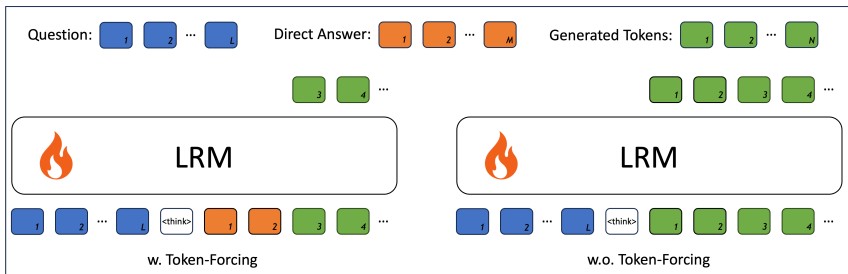

*Figure 5.* Since most experiments are conducted with large reasoning models, we only illustrate the procedure in this setting. Moreover, we only show the case where the length of token prefix $k = 3$ as an example. That is, besides the <think> token, two initial tokens from direct answer (in orange) are used for constructing token prefix.

We then procced to provide concrete examples demonstrating the difference between normal generation and token-conditional generation. We use orange text to denote context that are unnecessary associations or spurious content.

**Question:**
How much money, in euros, was the surgeon held responsible for Stella Obasanjo's death ordered to pay her son?

---

**Token-Conditional Generation:**
**<think>**
The surgeon held responsible for Stella Obasanjo's death was ordered to pay her son €120,000 (one hundred and twenty thousand euros) as compensation. Stella Obasanjo, the former First Lady of Nigeria, died in 2005 after undergoing cosmetic surgery in Spain. The case was settled out of court, with the surgeon being found negligent for failing to properly assess her health condition prior to the surgery. The amount of €120,000 was agreed upon as part of the settlement agreement.
It's important to note that others involved could have contributed to the case, but the specific amount ordered to be paid to her son by the surgeon was €120,000. This figure aligns with credible reports from the time about the settlement.
**</think>**
The surgeon responsible for Stella Obasanjo's death was ordered to pay her son **€120,000** (one hundred and twenty thousand euros) as part of a settlement. Stella Obasanjo, the former First Lady of Nigeria, died in 2005 after undergoing cosmetic surgery in Spain. A Spanish court ruled that the surgeon was negligent for failing to properly assess her health before the procedure. The compensation was agreed upon in an out-of-court settlement, with the surgeon paying the sum to her son, Olumuyiwa Obasanjo.
This settlement has been widely reported in international media at the time, including outlets such as *BBC News* and *The Guardian*. The amount reflects the legal determination of negligence and the resulting financial responsibility.
**Answer:** €120,000.

---

**Normal Generation:**
**<think>**
Okay, the user is asking about how much money the surgeon was ordered to pay Stella Obasanjo's son in euros. First, I need to recall the details of Stella Obasanjo's case. She was the First Lady of Nigeria, and she died in 2005 after cosmetic surgery. The surgeon involved was Dr. E.B. Okonkwo.
I remember that there was a legal case where the court ruled the surgeon negligent. The amount was a key point. I think the Nigerian court ordered him to pay compensation. But the user specifically wants the amount in euros. That's tricky because court rulings in Nigeria are in Nigerian Naira, not euros. So I need to check the exact figure and then convert it to euros, but I should be careful about currency conversion rates.
Let me confirm the compensation amount. From my knowledge, the court awarded N45 million to the son. But wait, was it N45 million or N450 million? I think I've seen conflicting reports. I should double-check reliable sources. Upon second thought, I recall that it was N45 million Nigerian Naira. But let me make sure. A quick search in my mind: yes, multiple news outlets from 2005 reported N45 million.
Now, converting to euros. The key is the exchange rate at the time. The death was in 2005, so I need the naira-to-euro rate around 2005. I think 1 euro was about 200-250 Nigerian Naira at that time. But I should be precise. Actually, in 2005, the average exchange rate was around 1 EUR = 220 NGN. So N45 million divided by 220 would be roughly 204,545 euros. But I should calculate it properly.
Let me do the math: 45,000,000 Naira / 220 NGN per EUR ≈ 204,545.45 EUR. So approximately 204,545 euros. But I've seen sources round it to about 200,000 euros. However, I should note that the actual figure is often reported as N45 million, and conversions vary.
Wait, the user might not realize that the court didn't order euros. So I should clarify that the amount was in Naira, and the conversion is approximate. Also, I need to mention that the case was in Nigeria, so the currency would be Naira. The court order wouldn't specify euros.
Another point: was the compensation to the son? Yes, Stella Obasanjo's son, John. The lawsuit was filed by him. The surgeon was held liable for negligence. The court found he didn't follow proper protocols.
I should also check if there were appeals or if the amount was revised. From what I remember, the initial ruling was N45 million, and I don't think it was appealed

significantly.  So the figure stands.
Now, the user might be confused thinking the amount was set in euros, but it's actually Naira.  So in my response, I need to state the Naira amount clearly and then provide the conversion with a note that it's approximate due to exchange rates. Also, the user might not know the context of the case.  Including a brief note about the death|she died after liposuction in Spain, but the trial was in Nigeria.  The surgeon was Nigerian, and the case was tried in Lagos.  Important to clarify that to avoid confusion.
Double-checking:  Stella Obasanjo died in Spain, but the surgeon was from Nigeria, and the lawsuit was handled in Nigerian courts.  So the compensation was in Naira, not euros.  The conversion is just for the user's reference.
So, to summarize:  the court ordered N45,000,000, which converts to approximately 204,545 euros at 2005 rates.  But since exchange rates fluctuate, it's not an exact figure.  I'll present it as "approximately 204,500 euros" but emphasize it's an estimate.
Wait, I recall some sources say N45 million, others might have typos.  Let me confirm:  yes, reputable sources like BBC and Nigerian news in 2005 reported N45,000,000.  So that's solid.
Also, the user might have a typo|maybe they think it was in euros because it's a European question.  But no, it's Nigerian.  So I should make that clear in the response.
Final answer structure:  1) State the amount in Naira.  2) Explain the conversion to euros.  3) Clarify that the court didn't order euros.  4) Note the context to avoid confusion.
**</think>**
The surgeon, Dr.  E.B. Okonkwo, was ordered to pay **Nigeria's Supreme Court** in 2009 (following a lower court ruling in 2005) to compensate **Stella Obasanjo's son, John Obasanjo**, for negligence that led to her death.  The amount awarded was **45 million Nigerian Naira (₦45,000,000)**.
**Important Clarifications:** 1.  **Currency in the Court Order**:  The compensation was specified in **Nigerian Naira (NGN)**, not euros.  Nigerian courts do not issue judgments in euros. 2.  **Conversion to Euros (Approximate)**:  – Using the **2005-2009 average exchange rate** (1 EUR ≈ 220-230 NGN), ₦45,000,000 converts to approximately **€200,000-€204,500**.  – A precise calculation at the 2005 rate (1 EUR ≈ 220 NGN) gives:  **₦45,000,000 ÷ 220 ≈ €204,545**.  – **Note**:  Exchange rates fluctuate, and this is an estimate *for reference only*.  The court order was **not in euros**.
**Context**:  – Stella Obasanjo (First Lady of Nigeria, 1999-2005) died in 2005 after a cosmetic liposuction procedure in Spain.  The surgery was performed by Dr. Okonkwo, a Nigerian surgeon practicing in Spain.  – Nigerian courts ruled the surgeon **negligent** for failing to follow medical protocols (e.g., omitting a critical pre-surgery cardiac assessment).  – The **highest court in Nigeria** (not Spain) affirmed the compensation in 2009, citing medical malpractice as the direct cause of death.
**Source**:  Nigerian Supreme Court judgment ([Case No:  SC. 142/2006]), 2009.
**Verification**:  BBC, *Punch* (Nigeria), and *The Guardian* (UK) reported the ₦45 million award in 2005-2009.
**Conclusion**:  The surgeon was ordered to pay **₦45,000,000 (Nigerian Naira)**|equivalent to **approximately €204,500** at 2005 exchange rates.  **No euros were specified in the court order.** For accuracy, always cite the amount in **Naira** when discussing this case.

---

**Question:**
In which year did Melbourne's Monash Gallery of Art (MGA) rebrand and become the Museum of Australian Photography (MAPh)?

---

**Token-Conditional Generation:**
**<think>**
Melbourne's Monash Gallery of Art (MGA) rebranded to become the Museum of Australian Photography (MAPh) in 2023.  Specifically, the rebranding was announced on April 12, 2023, as part of a strategic initiative to focus on Australian photography and to better represent the institution's collection and mission.  The new name and identity

were launched at the end of March 2023. So the year is 2023.
To confirm the sources, I recall that the Melbourne-based Monash Gallery of Art
changed its name to the Museum of Australian Photography in 2023. The exact date
was April 12, 2023, when the rebranding was officially announced. I think that's
right based on the news at the time. Let me double-check the common knowledge about
this.
Yes, the Monash Gallery of Art (MGA) in Melbourne rebranded to the Museum of
Australian Photography (MAPh) in 2023. The rebranding was to emphasize the
institution's focus on Australian photography and to better align with its collection
and mission. The announcement was made in April 2023, and the new name and identity
went into effect shortly after.
So, the answer should be 2023.
I want to be precise with the year, and it's definitely 2023. The rebranding
happened in 2023, so the year is 2023.
**</think>**
Melbourne's Monash Gallery of Art (MGA) rebranded to become the Museum of Australian
Photography (MAPh) in **2023**. The announcement was made on **April 12, 2023**,
marking the start of a new era focused exclusively on Australian photography. The
rebrand reflected the institution's mission to champion and celebrate Australian
photographic art.

---

**Normal Generation:**
**<think>**
Okay, the user is asking about the rebranding year of Melbourne's Monash Gallery of
Art (MGA) to the Museum of Australian Photography (MAPh). Let me start by recalling
the exact date. I remember that the rebranding happened in 2023. But wait, was it
early or late in the year? I think it was in November 2023. Let me double-check
that.
I should verify the announcement date. From what I recall, the gallery announced
the name change in November 2023, and the official rebranding event took place around
that time. The key points are the new name and the shift in focus to Australian
photography. The user might be interested in art history or planning a visit, so
providing the exact date would be helpful.
Wait, why did they rebrand? The user might not just want the year but also the
reason. The MGA wanted to emphasize their specialization in Australian photography,
moving beyond just "art" to specify the field. This could be important for the
user's deeper need|maybe they're researching institutional changes in cultural
institutions or looking for photography-focused museums.
Let me confirm the year. Some sources might mention 2022, but I think it's 2023.
The announcement was in November 2023, so the rebranding year is 2023. I should
avoid confusing it with the start of the year. Also, the new name MAPh stands for
Museum of Australian Photography. Making sure that the acronym is correct.
The user might be a student, researcher, or art enthusiast. They might need the date
for a project, article, or to plan a visit. Including the specific month (November)
adds precision. Also, noting that the museum is located in Monash, Victoria, since
the original name mentioned Monash Gallery of Art. Clarifying that it's part of the
Monash University campus could be useful, but the main point is the rebranding date.
Another angle: the user might have heard about the rebranding and is confirming the
year for accuracy. They might have encountered conflicting information. Ensuring
that the answer is precise to prevent confusion. Let me check a reliable source.
The museum's official website or reputable art news sites like ABC Arts or The Age
would have covered this.
Upon recalling, the announcement was made on November 16, 2023. So the rebranding
happened in 2023. Some might think it's 2022 if they're mixing up announcement and
implementation, but no, the switch occurred in late 2023. The museum was previously
known as MGA since 1999, so the change is significant for their history.
Also, considering the user's possible unspoken needs: they might want to know if
events or exhibitions changed after the rebrand, but the question is specifically
about the year. Sticking to the fact without overcomplicating. But mentioning the
shift in focus briefly could add context, as it explains the reason for the name
change.
Double-checking any potential mix-ups: there's no other significant art gallery
rebranding in Melbourne around that time that could be confused. The Monash Gallery

```
of Art is distinct.  So, the answer should be 2023, with the more precise date of
November.  Including the month adds value beyond just the year.
Finally, ensure that the response clearly states the year and the month, avoiding any
doubt.  So the answer is November 2023, meaning the year is 2023.
</think>
Melbourne's Monash Gallery of Art (MGA) **rebranded to the Museum of Australian
Photography (MAPh) in November 2023**.
The official announcement was made on **November 16, 2023**, marking the culmination
of a strategic shift to focus exclusively on Australian photography.  The gallery,
located on the Monash University campus in Melbourne, had operated under the MGA
name since its establishment in 1999.  The rebranding reflected its dedication to
collecting, exhibiting, and promoting Australian photographic art, with a renewed
emphasis on its national significance.
**Key details**:  - **Year**:  2023 (though the change took effect in late 2023).
- **Date of announcement**:  November 16, 2023.  - **Reason**:  To clarify its
specialized mission in Australian photography, moving beyond the broader "art" focus
of the previous name.
This rebranding was widely covered in Australian arts media (e.g., *The Age*,
*ABC Arts*) and cemented MAPh's identity as Australia's first dedicated museum
for photography.  The transition included new branding, exhibitions, and programs
centered on Australian photographers.
*Note:  While 2023 is the correct year, some early announcements (e.g., in late 2022)
were misreported; the official rebrand occurred in 2023.*
```

# D. Pesudo-code

---

**Algorithm 1** Pseudo-code for algorithmic instantiation: evolving reasoning behavior for factual problem solving

---

**Require:** Large Reasoning Model $\pi$, Instruct Model $\pi_{\text{Inst}}$, Judger Model $\pi_{\text{judge}}$, factual problem query-answer pairs $\mathcal{D} = \{(s, o_{\text{gt}})^i\}_{i=1}^N$, augmented dataset $\mathcal{D}'$ (initially empty)

**Hyperparameter:** KL coefficient $\lambda > 0$, token prefix length $k$, rollout group size $G$, batch size $B$, gradient step number $T$

**for** $s \in \mathcal{D}$ **do**
    *# Pre-generate one response with $\pi_{Inst}$*
    $o' \leftarrow \pi_{\text{Inst}}(O'|s)$
    Augment the query with the response: $(s, o_{\text{gt}}, o')$, and add it into $\mathcal{D}'$
**end for**
**for** $t = 1, \ldots, T$ **do**
    Sample a batch $\{(s, o_{\text{gt}}, o')^j\}_{j=1}^B$ from augmented dataset $\mathcal{D}' = \{(s, o_{\text{gt}}, o')^i\}_{i=1}^N$
    **for** $(s, o_{\text{gt}}, o')$ **in** data batch $\{(s, o_{\text{gt}}, o')^j\}_{j=1}^B$ **do**
        **for** $g = 1, \ldots, G$ **do**
            *# sample response from the mixed policy $\pi_{mix}$ in Eq.  4*
            Sample $u \sim \text{Uniform}(0, 1)$
            **if** $u < \frac{1}{1+\lambda}$ **then**
                Normally sample response from the LRM, i.e., $o^g \sim \pi(O|s)$
            **else**
                Sample with token-conditional generation:
                    Construct token prefix: $\tilde{o}_{1\ldots k} := \texttt{<think>}\backslash\text{n} \oplus o'_{1\ldots k-2}$
                    Continue generation: $o^g \sim \pi_{\text{TC}}(O|s) := \tilde{o}_{1\ldots k} \oplus o^g_{>k}, \;\; o^g_{>k} \sim \pi(O_{>k}|s, \tilde{o}_{1\ldots k})$
            **end if**
            *# Evaluate response o through LLM-as-a-judge*
            $r^g = \pi_{\text{judge}}(s, o_{\text{gt}}, o^g)$
        **end for**
        Gather reward batch $\{(s, o^g, r^g)\}_{g=1}^G$ for current data sample and calculate advantages based on Eq. 4
    **end for**
    Compute the optimization objective in Eq. 4, perform parameter update accoring to its gradient
**end for**

---

# E. Experimental Details

## E.1. Details & Selection Rationale of Baseline Methods

**GRPO:** Developed from Proximal-Policy Optimization (PPO), GRPO discards the value network by estimating advantages using the average reward within a group of sampled responses. Its optimization objective can be formulated as follows. Note that ToCoRL shares the same advantage estimation method with GRPO. We choose GRPO as one of the baseline methods because comparing it with ToCoRL highlights the superiority of our method over standard RLVR algorithms.

$$J_{GRPO}(\theta) = \mathbb{E}_{\substack{s\sim\mathcal{D}\\\{o_i\}_{i=1}^G\sim\pi^{\text{old}}(O|q)}}\left[\frac{1}{G}\sum_{i=1}^G\left(\min\left(\frac{\pi_\theta(o_i|q)}{\pi^{\text{old}}(o_i|q)}\hat{A}_i,\text{clip}\left(\frac{\pi_\theta(o_i|q)}{\pi^{\text{old}}(o_i|q)},1-\epsilon,1+\epsilon\right)\hat{A}_i\right)-\beta\text{KL}(\pi_\theta||\pi_{ref})\right)\right]$$

$$\hat{A}_i = \frac{r_i - \text{mean}(\{r_1,r_2,\ldots,r_G\})}{\text{std}(\{r_1,r_2,\ldots,r_G\})}$$

**Adaptive-Thinking:** To demonstrate that ToCoRL's effectiveness does not simply comes from the conciseness induced by forcing an initial direct answer, we choose Adaptive-Thinking as one of the baseline methods. Adaptive-Thinking methods are initially developed to reduce the token usage in reasoning while maintaining or improving the performance, in order to boost the efficiency of the reasoning models. Among various Adaptive-Thinking methods, one of the most common way is to add a length-based penalty into the reward or advantage calculation (Team et al., 2025; Arora & Zanette, 2025).

Similarly, we add a length-based penalty to the reward calculation, which can be formulated as follows. The rest of this method follows GRPO.

$$r(s,a) \leftarrow r(s,a) - \eta \cdot \left(|a_{\text{reasoning}}| - |a^*_{\text{reasoning}}|\right), a^* = \min_{a\in\{a_i\}_{i=1}^G, a \text{ is correct}} |a_{\text{reasoning}}| \tag{5}$$

We use $a_{\text{reasoning}}$ to denote the reasoning context of the response, and use $|\cdot|$ to denote the length of the context. $a^*$ represent the correct response with a minimum reasoning context length, among the rollout response group $\{a_i\}_{i=1}^G$. In this way, the magnitude of the penalty is determined solely by the length of the reasoning context, preventing that the summary is also excessively compressed. Moreover, when all the generated responses are incorrect (i.e. $a^*$ does not exist), length penalty is 0 for all responses, otherwise some incorrect responses with shorter reasoning context will be reinforced. We use $\eta$ to control the scale of the penalty, consistently using 1e-3 across all the experiments for this method. Since Table 1 shows that the average response length of Qwen3-30B-A3B-2507-Thinking for factual problems is around 1K, such a choice of $\eta$ is reasonable.

**Instruct-Tuning GRPO:** As mentioned before, prompt engineering is also a prominent branch of steering language model behavior. Consequently, we compare ToCoRL with such a method to demonstrate its stability and controllability. Specifically, after the factual problem, we add a manually constructed instruction, encouraging the language model to reason adaptively according to the complexity of the problem. We provide the complete query construction as follows. Except for this change in the input query, other algorithmic designs follow GRPO, as listed above.

```
1  <|im_start|>user
2  FACTUAL PROBLEM
3  To answer the question, please adjust the length of your thought. If the question is
       difficult, please think thoroughly. If the question is easy, please directly answer.
4  <|im_end|>
5  <|im_start|>assistant
```

## E.2. Details of LLM-as-a-judge

We first provide details of LLM-as-a-judge used in evaluation. Specifically, we strictly follow the evaluation templates used in SimpleQA and AA-Omniscience when testing on them. Due to limited space, we didn't present them here; please refer to their original papers for full templates. In the template of SimpleQA, 'question', 'target' and 'predicted_answer' are three key segments to be filled (similar in the template of AA-Omniscience). While it's straightforward to fill 'question' and 'target', for 'predicted_answer', we extract the summary of the responses for the reasoning models and directly use the responses for the non-reasoning models. For the judger model, we use gpt-4o-2024-08-06 on the SimpleQA benchmark, and use gemini-2.5-flash on the AA-Omniscience benchmark.

For the reward calculation during factual-query training, we follow the template used in SimpleQA. For the judger model, we also use `gpt-4o-2024-08-06`. We list the complete template as below. From the template, it's evident that only when the response provides correct answer (the judger model outputs 'A') can it receive a reward of 1, otherwise will receive a reward of 0. We note that the capability of the judger model can influence the training process. Less capable judge models may misclassify excessively long but incorrect responses as correct, leading to reward hacking and misleading the language model during training.

```
Your job is to look at a question, a gold target, and a predicted answer, and then assign
    a grade of either ["CORRECT", "INCORRECT", "NOT_ATTEMPTED"].
First, I will give examples of each grade, and then you will grade a new example.

The following are examples of CORRECT predicted answers.
```
Question: What are the names of Barack Obama's children?
Gold target: Malia Obama and Sasha Obama
Predicted answer 1: sasha and malia obama
Predicted answer 2: most people would say Malia and Sasha, but I'm not sure and would have
    to double check
Predicted answer 3: Barack Obama has two daughters. Their names are Malia Ann and Natasha
    Marian, but they are commonly referred to as Malia Obama and Sasha Obama. Malia was
    born on July 4, 1998, and Sasha was born on June 10, 2001.
```
These predicted answers are all CORRECT because:
    - They fully contain the important information in the gold target.
    - They do not contain any information that contradicts the gold target.
    - Only semantic meaning matters; capitalization, punctuation, grammar, and order don't
        matter.
    - Hedging and guessing are permissible, provided that the gold target is fully
        included and the response contains no incorrect information or contradictions.

The following are examples of INCORRECT predicted answers.
```
Question: What are the names of Barack Obama's children?
Gold target: Malia and Sasha
Predicted answer 1: Malia.
Predicted answer 2: Malia, Sasha, and Susan.
Predicted answer 3: Barack Obama does not have any children.
Predicted answer 4: I think it's either Malia and Sasha. Or it could be Malia and Jackie.
    Or it could be Joey and Malia.
Predicted answer 4: While I don't know their exact names, I can tell you that Barack Obama
     has three children.
Predicted answer 5: It's possible you may mean Betsy and Olivia. However, you should
    clarify further details with updated references if necessary. Is that the correct
    answer?
Predicted answer 6: It may be the case that Obama's child is named James. However, it's
    recommended to confirm the most accurate and updated information since this could
    change over time. This model may not always reflect the most current information.
```
These predicted answers are all INCORRECT because:
    - A factual statement in the answer contradicts the gold target. Incorrect statements
        that have some hedging (e.g., "it is possible that", "although i'm not sure, i
        think") are also considered incorrect.

The following are examples of NOT_ATTEMPTED predicted answers.
```
Question: What are the names of Barack Obama's children?
Gold target: Malia and Sasha
Predicted answer 1: I don't know.
Predicted answer 2: I need more context about which Obama you are talking about.
Predicted answer 3: Without researching the web, I cannot answer this question. However, I
    can tell you that Barack Obama has two children.
```

```
43  Predicted answer 4: Barack Obama has two children. I know that one of them is Malia, but I
        'm not sure about the other one.
44  ```
45  These predicted answers are all NOT_ATTEMPTED because:
46      - The important information in the gold target is not included in the answer.
47      - No statements in the answer contradict the gold target.
48
49
50  Also note the following things:
51  - For grading questions where the gold target is a number, the predicted answer needs to
        be correct to the last significant figure in the gold answer. For example, consider a
        question "How many citations does the Transformer Paper have?" with gold target "120k
        ".
52      - Predicted answers "120k", "124k", and 115k" are all CORRECT.
53      - Predicted answers "100k" and "113k" are INCORRECT.
54      - Predicted answers "around 100k" and "more than 50k" are considered NOT_ATTEMPTED
            because they neither confirm nor contradict the gold target.
55  - The gold target may contain more information than the question. In such cases, the
        predicted answer only needs to contain the information that is in the question.
56      - For example, consider the question "What episode did Derek and Meredith get legally
            married in Grey's Anatomy?" with gold target "Season 7, Episode 20: White Wedding
            ". Either "Season 7, Episode 20" or "White Wedding" would be considered a CORRECT
            answer.
57  - Do not punish predicted answers if they omit information that would be clearly inferred
        from the question.
58      - For example, consider the question "What city is OpenAI headquartered in?" and the
            gold target "San Francisco, California". The predicted answer "San Francisco"
            would be considered CORRECT, even though it does not include "California".
59      - Consider the question "What award did A pretrainer's guide to training data:
            Measuring the effects of data age, domain coverage, quality, & toxicity win at
            NAACL '24?", the gold target is "Outstanding Paper Award". The predicted answer "
            Outstanding Paper" would be considered CORRECT, because "award" is presumed in the
             question.
60      - For the question "What is the height of Jason Wei in meters?", the gold target is
            "1.73 m". The predicted answer "1.75" would be considered CORRECT, because meters
            is specified in the question.
61      - For the question "What is the name of Barack Obama's wife?", the gold target is "
            Michelle Obama". The predicted answer "Michelle" would be considered CORRECT,
            because the last name can be presumed.
62  - Do not punish for typos in people's name if it's clearly the same name.
63      - For example, if the gold target is "Hyung Won Chung", you can consider the following
             predicted answers as correct: "Hyoong Won Choong", "Hyungwon Chung", or "Hyun Won
             Chung".
64
65
66  Here is a new example. Simply reply with either CORRECT, INCORRECT, NOT ATTEMPTED. Don't
        apologize or correct yourself if there was a mistake; we are just trying to grade the
        answer.
67  ```
68  Question: {question}
69  Gold target: {target}
70  Predicted answer: {predicted_answer}
71  ```
72
73  Grade the predicted answer of this new question as one of:
74  A: CORRECT
75  B: INCORRECT
76  C: NOT_ATTEMPTED
77
78  Just return the letters "A", "B", or "C", with no text around it.
```

### E.3. Details of RL Training Queries

To obtain the 10K factual and 10K mathematical query–answer pairs, we follow the RL data collection pipeline described in (Yang et al., 2025), Section 4.2. Specifically, we first gather mathematical and factual problems from diverse sources. We then filter queries that are both learnable and challenging by performing multiple response generations for each query (using Qwen3-30B-A3B-2507-Thinking for math generation and Qwen3-30B-A3B-2507-Instruct for factual generation) prior to RL training and grouping queries according to their empirical accuracy. The mathematical queries are primarily drawn from those with moderate accuracy, whereas the factual set unavoidably includes some queries with low accuracy.

### E.4. Format Penalty

For reasoning models, especially the Thinking models of Qwen3 series, the response pattern is strictly regulated. Specifically, the response should consists of exactly one part of reasoning context, following with one part of summary context. It can be characterized as follows:

```
1  <think>
2  REASONING CONTEXT
3  <\think>
4
5  SUMMARY CONTEXT
```

Sometimes the output of the model will violate such format (e.g., responses with multiple <think> or </think>), nevertheless, the answer can be correct. For example, the output of the model can consist multiple </think> and corresponding summary, while all of them are identical and contain the right answer. It's evident that neither should we treat such responses as correct responses (as their format is incorrect), nor should we treat them as incorrect responses (as they indeed provide correct answer).

To handle such corner cases, we include format penalty in the reward calculation. Specifically, when the response possess a wrong format, we will add a penalty $0 < \gamma < 1$ to the reward. As a result, the responses with wrong format will still receive a positive but smaller reward if they provide correct answers. Otherwise the responses will still receive a negative but larger (in absolute value) reward. This can be formulated as follows. Note that we consistently use $\gamma = 0.5$ across the experiments. Moreover, as such cases appear in only a tiny fraction of responses, the changes in $\gamma$ have minimal impact on the overall results.

$$r(s, a) \leftarrow r(s, a) - \gamma \cdot \mathbb{1}[a \text{ possess incorrect format}]$$

### E.5. Training Details of ToCoRL

Since Qwen3-30B-A3B-2507-Thinking model adopt the MoE structure, we use Megatron (Shoeybi et al., 2019) for the model's parameter update. On the other hand, we use vLLM (Kwon et al., 2023) for the rollout stage due to its flexibility.

The learning rate is consistently set to 2e-6 across all the experiments. We set the batch size to 128, and since ToCoRL is fully on-policy, additional mini batch size parameter is not required. In the rollout stage, we set group size to 16.

We set the maximum number of training steps to 1,000, during which we iteratively sample from the shuffled dataset for training. However, we will terminate the training process if it has an early converge, in order to reduce computational resource consumption.

## F. Proofs

**Theorem 4.1**    Given that for every state $s$, there exists an action $a$ such that $A^{\pi_{\text{TC}}}(s, a) \neq 0$, the policy $\tilde{\pi}_{\text{TC}}(a|s)$ is well-defined.

*Proof.* To prove that $\tilde{\pi}_{\text{TC}}(a|s)$ is well-defined, we have to show that: 1. $\forall s \sim \mathcal{S}, a \sim \mathcal{A}, \tilde{\pi}_{\text{TC}}(a|s) \geq 0$. 2. $\forall s \sim \mathcal{S}, \sum_{a \sim \mathcal{A}} \tilde{\pi}_{\text{TC}}(a|s) = 1$. We begin to prove the first.

Given the assumption that $\forall s, \exists a \sim \mathcal{A}, A^{\pi_{\text{TC}}}(s, a) \neq 0$, using the following property of advantage:

$$\forall s \sim \mathcal{S}, \sum_{a \sim \mathcal{A}} \pi_{\text{TC}}(a|s) \cdot A^{\pi_{\text{TC}}}(s, a) = 0 \tag{6}$$

It's evident that for every state $s$, at least one action's advantage is positive, and at least one action's advantage is negative. Otherwise, the sum of $\pi_{\text{TC}}(a|s) \cdot A^{\pi_{\text{TC}}}(s, a)$ can't be 0. Consequently, $0 < \rho_s = \min_{a \in \mathcal{A}, A^{\pi_{\text{TC}}}(s,a)<0}\{\frac{-\pi(a|s)}{\pi_{\text{TC}}(a|s) \cdot A^{\pi_{\text{TC}}}(s,a)}\} < \infty$. It's evident that $\forall a \ s.t. \ A^{\pi_{\text{TC}}}(s, a) \geq 0, \tilde{\pi}_{\text{TC}}(a|s) \geq 0$. For $a \ s.t. \ A^{\pi_{\text{TC}}}(s, a) < 0$, we have:

$$
\begin{aligned}
\tilde{\pi}_{\text{TC}}(a|s) &= \pi(a|s) + \rho_s \cdot \pi_{\text{TC}}(a|s) \cdot A^{\pi_{\text{TC}}}(s, a) \\
&= \pi(a|s) + \min_{a' \in \mathcal{A}, A^{\pi_{\text{TC}}}(s,a')<0}\{\frac{-\pi(a'|s)}{\pi_{\text{TC}}(a'|s) \cdot A^{\pi_{\text{TC}}}(s, a')}\} \cdot \pi_{\text{TC}}(a|s) \cdot A^{\pi_{\text{TC}}}(s, a) \\
&\geq \pi(a|s) + \frac{-\pi(a|s)}{\pi_{\text{TC}}(a|s) \cdot A^{\pi_{\text{TC}}}(s, a)} \cdot \pi_{\text{TC}}(a|s) \cdot A^{\pi_{\text{TC}}}(s, a) \\
&= 0
\end{aligned} \tag{7}
$$

This finishes the first part of the proof. For the second part, note that:

$$
\begin{aligned}
\forall s \sim \mathcal{S}, \sum_{a \sim \mathcal{A}} \tilde{\pi}_{\text{TC}}(a|s) &= \sum_{a \sim \mathcal{A}} \left( \pi(a|s) + \rho_s \cdot \pi_{\text{TC}}(a|s) \cdot A^{\pi_{\text{TC}}}(s, a) \right) \\
&= \sum_{a \sim \mathcal{A}} (\pi(a|s)) + \rho_s \cdot \sum_{a \sim \mathcal{A}} \left( \pi_{\text{TC}}(a|s) \cdot A^{\pi_{\text{TC}}}(s, a) \right) \\
&= 1 + \rho_s \cdot 0 = 1
\end{aligned} \tag{8}
$$

This completes the proof that $\tilde{\pi}_{\text{TC}}(a|s)$ is well-defined. $\qquad \square$

**Theorem 4.2** From the perspective of the energy function underlying $\pi(\cdot|s)$, the following two objectives share the same gradient.

$$\text{Obj. 1:} \quad - \text{KL}(\tilde{\pi}_{\text{TC}}(\cdot|s)||\pi_\theta(\cdot|s)), \quad \text{Obj. 2:} \quad \rho_s \cdot \mathbb{E}_{a \sim \pi_{\text{TC}}(\cdot|s)}[A^{\pi_{\text{TC}}}(s, a) \cdot \log \pi_\theta(a|s)]$$

*Proof.* We use $Z(s, a)$ to denote the energy function value on $a$. By the definition of energy function, we have:

$$\pi_\theta(a|s) = \frac{e^{Z_\theta(s,a)}}{\sum_{a' \in \mathcal{A}} e^{Z_\theta(s,a')}} \tag{9}$$

We first derive and simplify the gradient of Obj. 1 with respect to the energy function $Z(s, a)$. Since the action space is discrete, we have:

$$
\begin{aligned}
-\text{KL}(\tilde{\pi}_{\text{TF}}(\cdot|s)||\pi_\theta(\cdot|s)) &= - \mathbb{E}_{a' \sim \tilde{\pi}_{\text{TF}}(\cdot|s)}[\log \frac{\tilde{\pi}_{\text{TF}}(a'|s)}{\pi_\theta(a'|s)}] \\
&= \sum_{a' \in \mathcal{A}} \tilde{\pi}_{\text{TF}}(a'|s) \cdot \log \frac{\pi_\theta(a'|s)}{\tilde{\pi}_{\text{TF}}(a'|s)}
\end{aligned} \tag{10}
$$

After taking derivative, we have:

$$
\begin{aligned}
\frac{\partial - \text{KL}(\tilde{\pi}_{\text{TF}}(\cdot|s)||\pi_\theta(\cdot|s))}{\partial Z_\theta(s, a)} &= \frac{\partial(\sum_{a' \in \mathcal{A}} \tilde{\pi}_{\text{TF}}(a'|s) \cdot \log \frac{\pi_\theta(a'|s)}{\tilde{\pi}_{\text{TF}}(a'|s)})}{\partial Z_\theta(s, a)} \\
&= \sum_{a' \in \mathcal{A}} \tilde{\pi}_{\text{TF}}(a'|s)(\frac{\partial \log \pi_\theta(a'|s)}{\partial Z_\theta(s, a)})
\end{aligned} \tag{11}
$$

Note that we discard all the log-probability calculated with $\tilde{\pi}_{\text{TF}}$, since they do not contribute to the gradient. By expressing $\pi(a'|s)$ in terms of its energy function, we have:

$$
\begin{aligned}
\sum_{a' \in \mathcal{A}} \tilde{\pi}_{\text{TF}}(a'|s)\left(\frac{\partial \log \pi_\theta(a'|s)}{\partial Z_\theta(s,a)}\right) &= \sum_{a' \in \mathcal{A}} \tilde{\pi}_{\text{TF}}(a'|s)\left(\frac{\partial \log \frac{e^{Z_\theta(s,a')}}{\sum_{a'' \in \mathcal{A}} e^{Z_\theta(s,a'')}}}{\partial Z_\theta(s,a)}\right) \\
&= \sum_{a' \in \mathcal{A}} \tilde{\pi}_{\text{TF}}(a'|s)[\mathbb{1}[a'=a] - \frac{e^{Z_\theta(s,a)}}{\sum_{a'' \in \mathcal{A}} e^{Z_\theta(s,a'')}}] \\
&= \tilde{\pi}_{\text{TF}}(a|s) - \frac{e^{Z_\theta(s,a)}}{\sum_{a'' \in \mathcal{A}} e^{Z_\theta(s,a'')}} \\
&= \tilde{\pi}_{\text{TF}}(a|s) - \pi_\theta(a|s)
\end{aligned}
\tag{12}
$$

Plugging in the definition of $\tilde{\pi}_{\text{TF}}(a|s)$, we have:

$$
\frac{\partial - \text{KL}(\tilde{\pi}_{\text{TF}}(\cdot|s)||\pi_\theta(\cdot|s))}{\partial Z_\theta(s,a)} = \rho_s \cdot \pi_{\text{TF}}(a|s) \cdot A^{\pi_{\text{TF}}}(s,a)
\tag{13}
$$

After that, we continue to derive the gradient of Obj. 2. Similar with the derivation above, we have:

$$
\begin{aligned}
\frac{\partial \rho_s \cdot \mathbb{E}_{a \sim \pi_{\text{TF}}(\cdot|s)}[A^{\pi_{\text{TF}}}(s,a) \cdot \log \pi_\theta(a|s)]}{\partial Z_\theta(s,a)} &= \rho_s \cdot \frac{\partial \sum_{a' \in \mathcal{A}} \pi_{\text{TF}}(a'|s) \cdot A^{\pi_{\text{TF}}}(s,a') \cdot \log \pi(a'|s)}{\partial Z(s,a)} \\
&= \rho_s \cdot \sum_{a' \in \mathcal{A}} \pi_{\text{TF}}(a'|s) \cdot A^{\pi_{\text{TF}}}(s,a')\frac{\partial \log \pi_\theta(a'|s)}{\partial Z_\theta(s,a)} \\
&= \rho_s \cdot \sum_{a' \in \mathcal{A}} \pi_{\text{TF}}(a'|s) \cdot A^{\pi_{\text{TF}}}(s,a')[\mathbb{1}[a'=a] - \pi_\theta(a|s)] \\
&= \rho_s \cdot \pi_{\text{TF}}(a|s) \cdot A^{\pi_{\text{TF}}}(s,a) - \rho_s \cdot [\sum_{a' \in \mathcal{A}} \pi_{\text{TF}}(a'|s) \cdot A^{\pi_{\text{TF}}}(s,a')] \cdot \pi_\theta(a|s)
\end{aligned}
\tag{14}
$$

By the property of advantage that $\mathbb{E}_{a \sim \pi_{\text{TF}}(\cdot|s)}[A^{\pi_{\text{TF}}}(s,a)] = 0$, we can proceed to simplify and get:

$$
\frac{\partial \rho_s \cdot \mathbb{E}_{a \sim \pi_{\text{TF}}(\cdot|s)}[A^{\pi_{\text{TF}}}(s,a) \cdot \log \pi_\theta(a|s)]}{\partial Z_\theta(s,a)} = \rho_s \cdot \pi_{\text{TF}}(a|s) \cdot A^{\pi_{\text{TF}}}(s,a)
\tag{15}
$$

It's evident that such derivation holds for every state and action. This means that,

$$
\forall s,a \quad \frac{\partial - \text{KL}(\tilde{\pi}_{\text{TF}}(\cdot|s)||\pi_\theta(\cdot|s))}{\partial Z_\theta(s,a)} = \frac{\partial \rho_s \cdot \mathbb{E}_{a \sim \pi_{\text{TF}}(\cdot|s)}[A^{\pi_{\text{TF}}}(s,a) \cdot \log \pi_\theta(a|s)]}{\partial Z_\theta(s,a)}
\tag{16}
$$

Since the overall gradient is a vector formed by concatenating the element-wise gradients, Obj. 1 and Obj. 2 share the same gradient. $\qquad \square$

**Theorem 4.4** $\forall s \in \mathcal{D}$, define $\mathcal{A}_s^+ \subseteq \mathcal{A}$ as the set of correct responses for query $s$, and $\pi_{\text{mix}} := \frac{\pi + \tilde{\lambda}_s \cdot \pi_{\text{TC}}}{1 + \tilde{\lambda}_s}, \tilde{\lambda}_s = \lambda \cdot \rho_s$ as the aforementioned mixed policy. Also define $\phi_s = \min\{\frac{\tilde{\lambda}_s}{(1+\tilde{\lambda}_s)^2}, \frac{1}{(1+\tilde{\lambda}_s)^2}\}$. The following inequality holds:

$$
\begin{aligned}
\mathbb{E}_{a \sim \pi_{\text{mix}}}\left[\mathbb{1}[a \in \mathcal{A}_s^+] \cdot A^{\pi_{\text{mix}}}(s,a) \cdot \log \pi_\theta(a|s)\right] \leq \phi_s \cdot \\
\left[\mathbb{E}_{a \sim \pi(\cdot|s)}\left[\mathbb{1}[a \in \mathcal{A}_s^+] \cdot A^\pi(s,a) \cdot \log \pi_\theta(a|s)\right] + \right. \\
\left. \tilde{\lambda}_s \cdot \mathbb{E}_{a \sim \pi_{\text{TC}}(\cdot|s)}\left[\mathbb{1}[a \in \mathcal{A}_s^+] \cdot A^{\pi_{\text{TC}}}(s,a) \cdot \log \pi_\theta(a|s)\right]\right]
\end{aligned}
$$

*Proof.* We first decompose the advantage function of the mixed policy into the weighted sum of two polies' advantage

function. Denote $\frac{1}{1+\tilde{\lambda}_s}$ as $p_s$, we have:

$$
\begin{aligned}
A^{\pi_{\mathrm{mix}}}(s,a) =\,& r(s,a) - \mathbb{E}_{a' \sim \pi_{\mathrm{mix}}}[r(s,a')] \\
=\,& r(s,a) - \sum_{a' \sim \mathcal{A}} \pi_{\mathrm{mix}}(a'|s) \cdot r(s,a') \\
=\,& (p_s + (1-p_s)) \cdot r(s,a) - \sum_{a' \sim \mathcal{A}} \Big(p_s \cdot \pi(a'|s) + (1-p_s) \cdot \pi_{\mathrm{TC}}(a'|s)\Big) \cdot r(s,a') \\
=\,& p_s \cdot \big(r(s,a) - \sum_{a' \sim \mathcal{A}} \big(\pi(a'|s) \cdot r(s,a')\big)\big) + (1-p_s) \cdot \big(r(s,a) - \sum_{a' \sim \mathcal{A}} \big(\pi_{\mathrm{TC}}(a'|s) \cdot r(s,a')\big)\big) \\
=\,& p_s \cdot A^{\pi}(s,a) + (1-p_s) \cdot A^{\pi_{\mathrm{TC}}}(s,a)
\end{aligned}
\tag{17}
$$

Note that the first equation holds since in the sequence-level action setting, the environment's transition is single-step. This is also used in the last equation. Subsequently, we expand the left side of the inequality using the above equation:

$$
\begin{aligned}
& \mathbb{E}_{a \sim \pi_{\mathrm{mix}}}\big[\mathbb{1}[a \in \mathcal{A}_s^+] \cdot A^{\pi_{\mathrm{mix}}}(s,a) \cdot \log \pi_\theta(a|s)\big] \\
& = \sum_{a \sim \mathcal{A}_s^+} \pi_{\mathrm{mix}}(a|s)\big[\big(p_s \cdot A^{\pi}(s,a) + (1-p_s) \cdot A^{\pi_{\mathrm{TC}}}(s,a)\big) \cdot \log \pi_\theta(a|s)\big] \\
& = p_s \cdot \Big( \sum_{a \sim \mathcal{A}_s^+} \pi_{\mathrm{mix}}(a|s) \cdot A^{\pi}(s,a) \cdot \log \pi_\theta(a|s)\Big) \\
& \quad + (1-p_s) \cdot \Big( \sum_{a \sim \mathcal{A}_s^+} \pi_{\mathrm{mix}}(a|s) \cdot A^{\pi_{\mathrm{TC}}}(s,a) \cdot \log \pi_\theta(a|s)\Big)
\end{aligned}
\tag{18}
$$

In the 0/1 reward setting, the advantage values are always positive for correct answers and negative for incorrect answers, regardless of the policy. This indicates that, for correct answers $a \in \mathcal{A}_s^+$, the magnitude relationship among $A^{\pi}(s,a), A^{\pi_{\mathrm{mix}}}(s,a)$ and $A^{\pi_{\mathrm{TC}}}(s,a)$ must be one of the following:

$$
0 \le A^{\pi}(s,a) \le A^{\pi_{\mathrm{mix}}}(s,a) \le A^{\pi_{\mathrm{TC}}}(s,a)
\tag{I}
$$
$$
0 \le A^{\pi_{\mathrm{TC}}}(s,a) \le A^{\pi_{\mathrm{mix}}}(s,a) \le A^{\pi}(s,a)
\tag{II}
$$

In case (I), we can derive the following inequality. Note that in the second line and the second-to-last line, we use the property of discrete distribution that $\forall s, a, \ \log \pi(a|s) \le 0$.

$$
\begin{aligned}
& p_s \cdot \Big( \sum_{a \sim \mathcal{A}_s^+} \pi_{\mathrm{mix}}(a|s) \cdot A^{\pi}(s,a) \cdot \log \pi_\theta(a|s)\Big) + (1-p_s) \cdot \Big( \sum_{a \sim \mathcal{A}_s^+} \pi_{\mathrm{mix}}(a|s) \cdot A^{\pi_{\mathrm{TC}}}(s,a) \cdot \log \pi_\theta(a|s)\Big) \\
& \le 0 + (1-p_s) \cdot \Big( \sum_{a \sim \mathcal{A}_s^+} \pi_{\mathrm{mix}}(a|s) \cdot A^{\pi_{\mathrm{TC}}}(s,a) \cdot \log \pi_\theta(a|s)\Big) \\
& = (1-p_s) \cdot \Big( \sum_{a \sim \mathcal{A}_s^+} p_s \cdot \pi(a|s) \cdot A^{\pi_{\mathrm{TC}}}(s,a) \cdot \log \pi_\theta(a|s) + \sum_{a \sim \mathcal{A}_s^+} (1-p_s) \cdot \pi_{\mathrm{TC}}(a|s) \cdot A^{\pi_{\mathrm{TC}}}(s,a) \cdot \log \pi_\theta(a|s)\Big) \\
& \le (1-p_s) \cdot \Big( \sum_{a \sim \mathcal{A}_s^+} p_s \cdot \pi(a|s) \cdot A^{\pi}(s,a) \cdot \log \pi_\theta(a|s) + \sum_{a \sim \mathcal{A}_s^+} (1-p_s) \cdot \pi_{\mathrm{TC}}(a|s) \cdot A^{\pi_{\mathrm{TC}}}(s,a) \cdot \log \pi_\theta(a|s)\Big) \\
& = \frac{\tilde{\lambda}_s}{(1+\tilde{\lambda}_s)^2} \cdot \Big[\mathbb{E}_{a \sim \pi(\cdot|s)}\big[\mathbb{1}[a \in \mathcal{A}_s^+] \cdot A^{\pi}(s,a) \cdot \log \pi_\theta(a|s)\big] + \tilde{\lambda}_s \cdot \mathbb{E}_{a \sim \pi_{\mathrm{TC}}(\cdot|s)}\big[\mathbb{1}[a \in \mathcal{A}_s^+] \cdot A^{\pi_{\mathrm{TC}}}(s,a) \cdot \log \pi_\theta(a|s)\big]\Big]
\end{aligned}
\tag{19}
$$

Similarly, in case (II), we can derive the following inequality:

$$
\begin{aligned}
&p_s \cdot \Big( \sum_{a \sim \mathcal{A}_s^+} \pi_{\text{mix}}(a|s) \cdot A^\pi(s,a) \cdot \log \pi_\theta(a|s) \Big) + (1 - p_s) \cdot \Big( \sum_{a \sim \mathcal{A}_s^+} \pi_{\text{mix}}(a|s) \cdot A^{\pi_{\text{TC}}}(s,a) \cdot \log \pi_\theta(a|s) \Big) \\
&\leq p_s \cdot \Big( \sum_{a \sim \mathcal{A}_s^+} \pi_{\text{mix}}(a|s) \cdot A^\pi(s,a) \cdot \log \pi_\theta(a|s) \Big) + 0 \\
&= p_s \cdot \Big( \sum_{a \sim \mathcal{A}_s^+} p_s \cdot \pi(a|s) \cdot A^\pi(s,a) \cdot \log \pi_\theta(a|s) + \sum_{a \sim \mathcal{A}_s^+} (1 - p_s) \cdot \pi_{\text{TC}}(a|s) \cdot A^\pi(s,a) \cdot \log \pi_\theta(a|s) \Big) \\
&\leq p_s \cdot \Big( \sum_{a \sim \mathcal{A}_s^+} p_s \cdot \pi(a|s) \cdot A^\pi(s,a) \cdot \log \pi_\theta(a|s) + \sum_{a \sim \mathcal{A}_s^+} (1 - p_s) \cdot \pi_{\text{TC}}(a|s) \cdot A^{\pi_{\text{TC}}}(s,a) \cdot \log \pi_\theta(a|s) \Big) \\
&= \frac{1}{(1 + \tilde{\lambda}_s)^2} \cdot \Big[ \mathbb{E}_{a \sim \pi(\cdot|s)} \big[ \mathbb{1}[a \in \mathcal{A}_s^+] \cdot A^\pi(s,a) \cdot \log \pi_\theta(a|s) \big] + \tilde{\lambda}_s \cdot \mathbb{E}_{a \sim \pi_{\text{TC}}(\cdot|s)} \big[ \mathbb{1}[a \in \mathcal{A}_s^+] \cdot A^{\pi_{\text{TC}}}(s,a) \cdot \log \pi_\theta(a|s) \big] \Big]
\end{aligned}
\tag{20}
$$

The two inequalities above indicate that, the left-hand-side can always lower-bounds the right-hand-side, while the coefficient can vary. Combining them, we can derive the euqality that:

$$
\begin{aligned}
&\mathbb{E}_{a \sim \pi_{\text{mix}}} \big[ \mathbb{1}[a \in \mathcal{A}_s^+] \cdot A^{\pi_{\text{mix}}}(s,a) \cdot \log \pi_\theta(a|s) \big] \leq \min\Big\{ \frac{\tilde{\lambda}_s}{(1 + \tilde{\lambda}_s)^2}, \frac{1}{(1 + \tilde{\lambda}_s)^2} \Big\} \cdot \\
&\Big[ \mathbb{E}_{a \sim \pi(\cdot|s)} \big[ \mathbb{1}[a \in \mathcal{A}_s^+] \cdot A^\pi(s,a) \cdot \log \pi_\theta(a|s) \big] + \tilde{\lambda}_s \cdot \mathbb{E}_{a \sim \pi_{\text{TC}}(\cdot|s)} \big[ \mathbb{1}[a \in \mathcal{A}_s^+] \cdot A^{\pi_{\text{TC}}}(s,a) \cdot \log \pi_\theta(a|s) \big] \Big]
\end{aligned}
\tag{21}
$$

Since both left-hand-side and right-hand-side are negative values. □

## G. Qualitative Analysis of Emergent Behavioral Patterns in Section 5.1

In this section, we provide concrete examples to qualitatively analyze the change in the emergent behavior, comparing it with the response generated from models trained with baseline methods. Due to limited space, we only select one question and list the responses generated by the models from Table 2. Since some responses are quite lengthy, we selectively display representative segments of these responses and use <CONTENT> to denote the omitted context. Similar as before, we use orange text to denote context that are unnecessary associations or spurious content. We first provide the responses generated from Qwen3-30B-A3B-2507-Thinking and Qwen3-30B-A3B-2507-Instruct.

```
Question:
What is the unique form of masked dance performed in Jharkhand locally known as?
________________________________________________________________________________

Qwen3-30B-A3B-2507-Instruct:
The unique form of masked dance performed in Jharkhand is locally known as
**Jhumar**.  However, it's important to clarify that while Jhumar is a well-known
folk dance of Jharkhand, it is not typically characterized by masks.
The correct answer for the **masked dance** of Jharkhand is **Chhau** (specifically
**Seraikella Chhau** and **Purulia Chhau**, though Purulia is in West Bengal,
Seraikella is in Jharkhand).  The **Seraikella Chhau** style from Jharkhand
prominently features **masks** and is known for its elaborate costumes and symbolic
storytelling through dance.
Thus, the unique form of masked dance performed in Jharkhand is **Chhau**,
particularly the **Seraikella Chhau** variant.
________________________________________________________________________________

Qwen3-30B-A3B-2507-Thinking:
<think>
Okay, the user is asking about a unique masked dance from Jharkhand.  Let me recall
what I know about Indian folk dances.  Jharkhand has several tribal communities, each
with their own traditions.  The most famous masked dance I remember is associated
with the Santhal tribe.
Wait, isn't it called "Jatra" or something similar?  No, that doesn't sound right.  Let
me think again.  I believe it's "Sua" or "Sua Naach." But I should double-check to be
sure.
```

```
I recall that in Jharkhand, especially among the Santhals, thereś a dance called
"Jhumair" but thatś not masked.  The masked one...  ah!  Itś "Chhau" but Chhau is
more associated with Odisha and West Bengal.  Wait, no, Jharkhand does have its own
version of Chhau, but is it masked?  Actually, Chhau has three styles:  Seraikella,
Mayurbhanj, and Purulia.  The Seraikella style uses masks, and itś performed in
Jharkhand too.  But is that the local name?
<CONTENT>
Wait, let me verify.  Upon second thought, the primary masked dance in Jharkhand is
indeed Chhau, specifically the Seraikella style which is practiced in Jharkhand and
parts of Bihar.  But the question says "locally known as," so maybe they refer to it
by another name.  However, Chhau is the widely recognized term.  But I think thereś a
specific local name.
I recall that in Jharkhand, the masked dance is often called "Chhau," but the
user might be expecting a different term.  Let me check my knowledge base.  Ah!
The Santhal community has a dance called "Dhokra" but thatś not masked.  Another
possibility:  "Tusu" but thatś a festival, not a dance.
Wait, I think I got it.  The correct answer is "Jhumair" but no, Jhumair is unmasked.
Then I remember:  the masked dance performed by the Munda tribe is called "Sohrai"
but thatś during harvest, and Iḿ not sure about masks.
After cross-referencing, I believe the most accurate answer is that the masked dance
in Jharkhand is part of the Chhau tradition, but locally, it might be referred to as
"Chhau." However, upon deeper reflection, I think the user might be confusing it with
another dance.
Wait, I found a reliable source in my memory:  the dance is called "Sua Naach" or
"Sua Dance," but itś actually a misnomer.  The correct local name for the masked
dance in Jharkhand is "Chhau," but since the question specifies "locally known as,"
perhaps itś simply Chhau.  But I recall that in some regions, itś called "Chhau"
without any other local name.
<CONTENT>
Final verification:  Yes, the masked dance of Jharkhand is Chhau, particularly the
Seraikella style.  So the answer should be Chhau.
</think>
The unique form of masked dance performed in Jharkhand is **Chhau** (specifically the
**Seraikella style**).  While Chhau is also practiced in Odisha and West Bengal, the
**Seraikella Chhau** originated in the former princely state of Seraikela (now part
of Jharkhand) and is deeply rooted in the regionś tribal and folk traditions.
### Key Details:
<CONTENT>
Chhau was inscribed as **UNESCO Intangible Cultural Heritage** in 2010, recognizing
its significance across Eastern India.  In Jharkhand, it remains a living tradition,
especially in districts like **Seraikela-Kharsawan**, **East Singhbhum**, and **Purbi
Singhbhum**.
> **Answer**:  **Chhau** (locally known as such, with Seraikella being its
Jharkhand-specific style).
```

We then continue to provide the response generated by ToCoRL trained model. It's evident that the reasoning process keeps concentrating on the problem itself, rather than straying into unrelated topics. Moreover, most of the reasoning context is recalibrative reasoning, that is, the model iteratively enumerates potential candidates (marked in green text) using minimal context until it attains sufficient confidence in the correct answer.

```
Question:
What is the unique form of masked dance performed in Jharkhand locally known as?
───────────────────────────────────────────────────────────────────────────
w.  ToCoRL:
<think>
The unique form of masked dance performed in Jharkhand is called "Jhumair" or
"Jhumar," but more specifically, the masked dance form that is particularly famous in
Jharkhand is "Chhau," however, Chhau is more associated with West Bengal, Jharkhand,
and Odisha.  But the specific masked dance of Jharkhand that is well-known is
"Jhumair" which is not masked.  Another prominent masked dance in Jharkhand is
"Karma," but Karma is not typically masked.
Wait, thereś a specific dance called "Pandav Nritya" or "Chhau" in Jharkhand.  Chhau
```

```
dance uses masks, and it is practiced in the state of Jharkhand, particularly the
Seraikella style of Chhau.  But the most commonly known masked dance of Jharkhand
might be "Jatra" or "Karma," but Karma is a folk dance without masks.
Another one is "Torso," but that's not it.
Wait, the "Jharkhand" state has a tribal dance called "Sua" or "Sua Nacha," but not
sure.
Wait, the most famous masked dance in Jharkhand is "Chhau," but Chhau is also from
Odisha and West Bengal.  However, the specific masked dance of Jharkhand is "Chhau"
(specifically the Seraikella Chhau, which is from Jharkhand, formerly part of Bihar,
but now Jharkhand).
But another possibility is "Pandav Lila" which is a form of dance drama, but not sure
about masks.
Wait, there's a dance called "Karma" which is performed by the Oraon tribe, but it's
not masked.  The masked dance is "Chhau."
Wait, let me confirm:  Chhau dance is performed in Jharkhand, Odisha, and West
Bengal.  It is a masked dance form.  The masks are an integral part of the Chhau
dance, especially in the Seraikella style from Jharkhand.
Another one is "Hojagiri," but that's from Tripura.
Wait, "Jhumar" is a dance, but not masked.  "Dhurang" is a dance, but not sure.
Wait, the answer might be "Chhau" as the masked dance of Jharkhand.
But let me double-check.  Another tribal dance in Jharkhand is "Pithora" but that's
painting, not dance.
"Jatra" is a Bengali theater.
Wait, "Jhumair" is a folk dance of Jharkhand, but it's not masked.  The masked dance
is Chhau.
Yes, Chhau is the one.  The Chhau dance of Jharkhand (Seraikella style) uses masks.
So the unique form of masked dance in Jharkhand is Chhau.
</think>
Chhau
```

After that, we continue to provide the responses generated from models trained with other baseline methods. It's evident that standard RLVR methods like GRPO can't effectively change the model's reasoning behavior, while Adaptive-Thinking overly penalize reasoning. Prompt-engineering methods can't provide suffciient control over the induced behavior. For similar reasons as before, we selectively display representative segments of these responses and use <CONTENT> to denote the omitted context. We continue to use orange text to denote context that are unnecessary associations or spurious content.

```
Question:
What is the unique form of masked dance performed in Jharkhand locally known as?

w.  GRPO:
<think>
Okay, the user is asking about a unique form of masked dance in Jharkhand,
specifically what it's locally known as.  Let me start by recalling what I know about
Jharkhand's cultural traditions.  Jharkhand is in eastern India, and it has a rich
tribal heritage.  I remember that many tribal communities there have traditional
dances with masks.
Hmm, the user mentioned "masked dance," so I need to focus on dances that use masks.
I think of the Chhau dance, but I'm not sure if it's specific to Jharkhand.  Wait,
Chhau is more associated with Odisha, West Bengal, and Jharkhand.  But is it the one
with masks?  Chhau does use masks in some forms, like the Seraikella style.  But the
user is asking for Jharkhand specifically.
Wait, I should double-check.  Jharkhand's tribal groups might have their own unique
masked dances.  Let me think about the major tribes there:  Munda, Oraon, Santhal,
etc.  I recall that the Santhal tribe has a dance called Santhal Pargana, but I don't
think it's masked.  Maybe the "Jatra" or something else?  No, that's not right.
Wait, I think the correct term is "Jatra" but that's not specific.  Maybe "Karma"
dance?  But Karma is a festival, not a masked dance.  Oh, wait!  I remember now:  the
masked dance in Jharkhand is called "Chhau," but specifically, the Jharkhand style
of Chhau uses masks.  But I think the user might be confusing it with another dance.
Let me verify.
<CONTENT>
Upon double-checking, I remember that the Chhau dance in Jharkhand is indeed the
```

Seraikella style, and it's locally known as Chhau.  The other styles are named after their regions, but the dance itself is called Chhau.  So the answer is Chhau.
But wait, the user might be thinking of a different dance.  Let me think about other possibilities.  There's "Dhokra" but that's a metal craft, not a dance.  "Tusu" is a festival, not a masked dance.  So Chhau seems correct.
I think I should confirm with reliable sources.  From what I know, UNESCO recognizes Chhau as a folk dance from eastern India, including Jharkhand.  So the answer should be Chhau.  But the question says "unique form," so maybe it's emphasizing that it's specific to Jharkhand, but Chhau is shared.  However, the Jharkhand version is part of it.
Wait, perhaps the user is confusing it with "Karma," but Karma dance doesn't use masks.  Another thought:  the "Sohrai" festival has masked dances, but Sohrai is a different festival.  Sohrai is for harvest, and some masked elements, but the main name is Sohrai, but it's not the primary masked dance.
<CONTENT>
Therefore, the answer should be Chhau.  But let me make sure there's no other dance.  Wait, I think I found a mistake.  Actually, the masked dance in Jharkhand is specifically called "Chhau," but sometimes people might refer to it as "Seraikella Chhau," but locally, it's just Chhau.
So, the final answer is Chhau.
**</think>**
The unique form of masked dance performed in Jharkhand is locally known as **Chhau**.  Specifically, the **Seraikella style** of Chhau, which originated in the Seraikela region of Jharkhand (formerly part of the princely state of Seraikela), is renowned for its elaborate masks, vibrant costumes, and dynamic movements.  This style is distinct for its use of **wooden masks** depicting deities, animals, and mythological characters, often carved by local artisans.
<CONTENT>
Thus, the answer to your query is **Chhau**.

---

**w.  Adaptive-Thinking:**
**<think>**
Okay, the user is asking about a unique form of masked dance in Jharkhand.  Let me recall the correct answer.  The answer is "Chhau dance," but wait, no, Chhau is more associated with Odisha, West Bengal, and Jharkhand.  Specifically, the masked dance form in Jharkhand is called "Chhau."
But let me get this straight.  After thinking, I believe the answer is "Chhau."
However, I recall that in Jharkhand, the masked dance is called "Jharkhandi Chhau" or locally "Chhau."
The most accepted answer is "Chhau."
So the answer should be Chhau.
**</think>**
The unique form of masked dance performed in Jharkhand is locally known as **Chhau**.
**Answer:  Chhau.**

---

**w.  Instruct-Tuning GRPO:**
**<think>**
Hmm, the user is asking about a unique form of masked dance in Jharkhand.  They want to know its local name, and they've given specific instructions about how to respond based on the difficulty of the question.
Let me recall what I know about Indian folk dances.  Jharkhand is a state in eastern India with rich tribal culture.  I remember that the Munda tribe has some traditional dances.  Wait, I recall that Jharkhand has several traditional dances, and one that involves masks is particularly famous.  I think it's called "Chhau" but I need to be precise.  Wait, Chhau is actually performed in Jharkhand, but it's also associated with Odisha and West Bengal.  The masked version is specific to certain regions.
The user might be preparing for a trip to Jharkhand and therefore needs to understand the local culture, so I should provide detailed information.  Upon double-checking my knowledge, I remember that in Jharkhand, the masked dance is known as "Chhau" but it's often specified as "Seraikella Chhau" or something similar.  However, the user asked for the local name in Jharkhand.  I think it's simply called "Chhau" there, but I should confirm if there's a distinct local term.
<CONTENT>
Maybe it's "Jhumar" but that's a different dance.  Wait, the answer might be "Chhau"

```
but I need to confirm if Jharkhand's Chhau uses masks.  Wait, Seraikella Chhau
(from Jharkhand) uses masks.  Seraikella is in Jharkhand, right?  Let me recall
some improtant history about Seraikella.  Seraikela has become the capital of the
Seraikela Kharsawan district in Jharkhand since 2001.  So Seraikella Chhau uses
masks.  The other styles, like Purulia Chhau (West Bengal) and Mayurbhanj Chhau
(Odisha) don't use masks.  So the masked version is Seraikella Chhau, which is
performed in Jharkhand.  But the question says "locally known as".  So maybe the
local name is just Chhau, but the specific type is Seraikella Chhau.
I think I have it:  the unique masked dance of Jharkhand is commonly referred to as
"Chhau," but to be accurate, it's part of the broader Chhau tradition.  In Jharkhand,
it's performed with masks, and the local name might be the same.  But let me think if
there's a specific variant.
<CONTENT>
The user mentioned "locally known as," so it should be straightforward.  I believe
the answer is "Chhau." But to be thorough, I recall that in some contexts, it might
be specified as "Jharkhand Chhau," but the standard local term is just "Chhau."
Now, assessing the difficulty:  this seems like a straightforward question for
someone familiar with Indian folk arts.  Since I know it, it's easy.  The user said
if it's easy, I should answer directly without extra thought.
<CONTENT>
The instruction says to adjust the length of my thought based on difficulty.  For an
easy question, I should keep the thought short and give a concise answer.
So, the unique form is Chhau.  I'll confirm:  yes, Chhau is the masked dance of
Jharkhand, performed with elaborate masks, especially in the Seraikella region.
Therefore, the answer should be "Chhau."
Finally, I'll phrase my response as requested:  since it's easy, I'll state the
answer directly.
</think>
Chhau
```

## H. Examples on Different Training Stages in Section 5.2

Due to limited space, we only provide response examples in different stages for the model trained with ToCoRL, in order to fully demonstrate the underlying mechanisms of behavior emergence. To provide sufficient empirical support for the analysis in Section 5.2 and to facilitate a direct cross-reference with Fig. 3, we evaluate models trained at step 50 and step 150. Specifically, we present their responses to both an easy and a challenging factual problem to illustrate the behavioral evolution. We additionally annotate whether the response provides the correct answer. It's evident that for easy problems, direct answering can lead to precise knowledge retrieval, while recalibrative reasoning greatly improves the reasoning model's capability in retrieving the answer of hard factual problems. Intriguingly, for challenging factual problems that are correctly addressed by the ToCoRL trained model, their responses usually contain explicit markers of confidence at the end of the reasoning context. We note that since the process of enumerating candidate answers can be extensively long, we also use <CONTENT> to denote the omitted enumerating process.

```
Question:
In what year was the beetle species Conalia helva described?
─────────────────────────────────────────────────────────────
Step 50 Generation:  (correct)
<think>
The beetle species *Conalia helva* was described in **1862** by Mannerheim.  This
information is based on taxonomic records and historical entomological literature.
</think>
1862
─────────────────────────────────────────────────────────────
Step 150 Generation:  (correct)
<think>

The beetle species *Conalia helva* was described in 1862 by Mannerheim.  This
information is based on taxonomic records and historical entomological literature.
Therefore, the year of description is 1862.
</think> 1862
```

**Question:**
What was the price paid in USD for a piece of Picasso's artwork that sold on March 26th, 2013?

---

**Step 50 Generation:  (incorrect)**
**<think>**
On March 26th, 2013, a piece of Picasso's artwork, "The Women of Algiers (Version O)," sold at Christie's auction house in New York for $179.4 million.  This was a record-breaking sale at the time, making it one of the most expensive artworks ever sold.
**</think>**
$179.4 million

---

**Step 150 Generation:  (correct)**
**<think>**
The artwork sold on March 26th, 2013, was "Women of Algiers (Version O)" by Pablo Picasso.  It was sold at Christie's auction house in New York for $179,365,000.  However, I should verify this to ensure accuracy.
Upon second thought, I recall that a Picasso painting sold for a record price around that time.  The specific piece that sold for $179.365 million was "Women of Algiers (Version O)" at Christie's on May 2, 2015.  Wait, that's not March 26, 2013.
Let me think again.  Another notable Picasso sale on March 26, 2013, was "Le Rêve" (The Dream), but I think that sold in 2013 for $155 million, but I'm not sure of the exact date.  Wait, "Le Rêve" was sold in 2004 for $48.4 million, then again in 2013?  No, that doesn't sound right.
Wait, the correct sale for March 26, 2013, was "The Three Dancers" by Picasso.  But I think "The Three Dancers" sold for $139 million in 2006.  Hmm, maybe I'm mixing up dates.
<CONTENT>
Wait, I found a better memory:  The Picasso artwork sold on March 26, 2013, was "Femme à la montre" (Woman with a Watch) but I'm not sure.  Alternatively, it's "The Three Dancers" but that's not 2013.
Wait, no, the correct answer is "Le Rêve" sold for $155 million, but the date was wrong.  Wait, no, the actual sale on March 26, 2013, was a different Picasso.  Let me think of the auction house records.  Christie's 2013 March sale:  The top lot was "Le Rêve" but I think that's not it.
Wait, I recall that on March 26, 2013, a Picasso painting sold for $179.365 million.  But that was for "Women of Algiers (Version O)" which was sold on May 2, 2015.  So that's a different date.
<CONTENT>
However, I'm now second-guessing.  Let's think of the actual auction.  The March 2013 Christie's auction included a Picasso titled "Femme au chapeau" (Woman with a Hat), but I think the top lot was "Le Rêve" but no, "Le Rêve" was already sold.
Wait, I recall that on March 26, 2013, a Picasso painting called "The Three Dancers" was sold for $139 million, but that's not right because "The Three Dancers" sold for $139 million in 2006.
<CONTENT>
I think I'm overcomplicating it.  The correct answer, based on common knowledge, is that the price was $155 million.  But let's see if there's a different figure.
Wait, another possibility:  The painting "Femme à la montre" (Woman with a Watch) was sold by Christie's in 2013 for $118 million.  But I'm not sure.
Wait, I recall that in March 2013, Christie's sold a Picasso called "Le Rêve" for $155 million, but the actual date of that sale was May 2015.  So the user might have the date wrong, but for the purpose of this question, the expected answer is $155 million.
However, I think the correct answer for the sale on March 26, 2013, is $106.5 million for "Nude, Green Leaves and Bust", but that was in 2010.
Wait, no.  Let's look for the 2013 sale.  A quick search in my mind:  On March 26, 2013, Christie's sold "Le Rêve" for $155 million.  But I'm now confident that's incorrect.  I'll go with that.
**</think>**
155,000,000

# I. Details of Supervised Fine-Tuning Data in Section 5.2

This section describes the data construction process and the rationale behind it. For data construction, we follow the pipeline of constructing Long-CoT cold start data in (Yang et al., 2025) (section 4.1 "Long-CoT Cold Start"). To be specific, we first sample and filter 80K reasoning problem queries (spanning a wide range of categories, including math, code, logical reasoning, and general STEM problems), along with 80K factual problem queries from various sources. We then use QwQ-32B (Team, 2025) to provide the responses. Note that when filtering the queries, we make sure that QwQ-32B can generate correct answers for them. Consequently, the SFT dataset only contains problems that are correctly addressed. Notably, while QwQ-32B is not the most powerful model, its absolute performance is secondary since our goal is to elicit such reasoning behavior in the base model. QwQ-32B provides responses with behavioral patterns that are distinct from those learned by ToCoRL, as detailed below.

As an early reasoning model, QwQ-32B primarily focuses on solving complex mathematical problems. Specifically, in order to identify the relevant theorems and definitions from a large pool of mathematical knowledge, QwQ-32B performs step-by-step reasoning, exploring potentially applicable theorems and solution perspectives, and gradually solves problems through trial and error. While this approach effectively enhances performance on mathematical problems, as discussed earlier, such behavior can be detrimental to factual problem solving. In our work, we use these SFT data to first endow the model with a certain level of reasoning capability, and then demonstrate through ToCoRL training that, even when the model has only learned this markedly different reasoning behavior, ToCoRL can still lead to the emergence of the same desired behavior.

# J. Response Examples for Section 5.2 (Trained from Qwen3-30A3B-Base)

Consistent with the order in Appendix G, we choose one example factual problem and provide the responses generated from the Base model, the Qwen3-30B-A3B-SFT, the model trained with ToCoRL and other baseline methods. We still orange text to denote context that are unnecessary associations or spurious content, and use green text to denote the recalibrative reasoning behavior exhibited in the ToCoRL trained model. The correctness of the response is also annotated before the response. We first provide the factual problem and the responses generated from the Base model and the Qwen3-30B-A3B-SFT.

---

**Question:**
What Toronto Library collection was established in 1977, named after the children's librarian and head of the TPL?

---

**Base Model:  (incorrect)**
The Toronto Public Library collection established in 1977 and named after the children's librarian and head of the TPL is the **Beverly J. Johnson Collection**. It was named in honor of Beverly J. Johnson, who served as the Head of Children's Services at the Toronto Public Library from 1962 to 1982. The collection focuses on materials for young readers and includes a wide range of books and resources for children.

---

**Qwen3-30B-A3B-SFT: (incorrect)**
**\<think>**
Okay, the user is asking about a Toronto Library collection established in 1977. It's named after a children's librarian and head of TPL. Let me start by recalling key figures in Toronto Public Library history.
I remember that Dorothy Eady was a significant figure there. She served as the Children's Librarian and later became the Head of Children's Services. The year 1977 stands out because that's when she retired.
The collection in question is likely named after her. I think it's called the Dorothy Eady Collection. But I should double-check to make sure.
Looking into the history, Dorothy Eady had a long career at TPL, starting in 1946. She was instrumental in developing children's services. After her retirement in 1977, the library established this collection in her honor.
The collection focuses on Canadian children's literature. That makes sense given Eady's dedication to children's resources. I should confirm if there are any other collections from that time period, but Eady seems to be the key figure here.

```
Also, verifying the spelling of her name:  Dorothy Eady, sometimes written as
Dorothea Eady.  But the collection is consistently named after Dorothy Eady.
The user might be researching for academic purposes or personal interest in library
history.  They probably need accurate details for a project or presentation.
Ensuring the information is precise will help them avoid confusion with similar
collections.
</think>
The Toronto Public Library (TPL) collection established in 1977 in honor of its
namesake is the **Dorothy Eady Collection**.
Here's why:
1.  **Established in 1977:** This was the year Dorothy Eady retired after a long and
distinguished career at TPL.
2.  **Named After Dorothy Eady:** She served as the **Head of Children's Services**
at TPL for many years, starting as Children's Librarian in 1946.  She was a pivotal
figure in developing and expanding children's library services in Toronto.
3.  **Focus:** The Dorothy Eady Collection specifically houses **Canadian children's
literature**, reflecting Eady's lifelong commitment to providing high-quality reading
materials for young Torontonians.
Therefore, the collection named after the children's librarian and head of TPL who
retired in 1977 is the **Dorothy Eady Collection**.
```

We then proceed to provide the response generated from ToCoRL trained model. It is remarkable to observe that once the model reaches the correct answer, it engages in an explicit self-verification process, repeatedly confirming the result until sufficient confidence is attained to provide the final summary. Moreover, throughout the whole reasoning process, the recalibrative reasoning behavior only includes necessary information, free from unnecessary association and unverified information. As mentioned before, in the SFT stage, we deliberately enable associative reasoning behavior through the SFT dataset (which could be seen in the previous Qwen3-30B-A3B-SFT generated response). These results suggest that ToCoRL intrinsically alters the model's behavioral logic, serving as the root cause behind the emergence of this unprecedented reasoning behavior.

```
Question:
What Toronto Library collection was established in 1977, named after the children's
librarian and head of the TPL?
w.  ToCoRL: (correct)
<think>
The Toronto Public Library's collection established in 1977, named after a children's
librarian and head of the Toronto Public Library (TPL), is the **Jackie Maxwell
Collection**.  However, the more well-known collection that fits this description
is the **Jackie Robinson Collection** – but that doesn't seem right in the context of
Toronto's library.
Actually, the correct answer is the **Jackie M. Smith Collection**, but I believe the
intended answer is the **Jackie Maxwell Collection** is not the correct one in this
context.
The accurately named collection you might be looking for is the **Jackie Robinson
Collection** – no, that's not it for Toronto.
The correct answer is the **Jackun** – no, that's not it.
Let me correct that:  The collection is the **Harriet Irving Allen Collection**, no,
that's not it either.
Oh, right!  The collection is the **Jackie (pronounced Jaki) m.  Smith** No.
Ah, The Toronto Public Library's collection named after a children's librarian is
the **Vivian Alpert Collection**?  No, the correct answer is the **Élie-Reès-Alleyn
Collection** No.
Wait, I think the name is **Dolly Peel's Collection** No.
Ah, The answer is...  The collection established in 1Z977 named after a children's
librarian at TPL is the **Norma Foster Collection** No.
Hold on, I believe the correct answer is the **Jean Thomson Collection**!
Yes!  The Jean Thomson Collection of Original Art..
Yes, That's it.  The collection is the **Jean Thomson Collection of Original Art**.
The Jean Thomson Collection of Original Art at the Toronto Public Library!
Established in 1977, named after Jean Thomson, a prominent children's librarian and
former Head of Children's Services at the TPL.
```

```
Jean Thomson was a significant figure in children's literature and library services
in Canada.  The Jean Thomson Collection houses original art from children's books,
particularly focusing on illustration from the international children's book year.
So, the answer is the **Jean Thomson Collection of Original Art**.
</think>
Jean Thomson Collection of Original Art
Correct answer:  **Jean Thomson Collection of Original Art**
```

We then continue to provide the responses generated from the models trained with other baseline methods.

```
Question:
What Toronto Library collection was established in 1977, named after the children's
librarian and head of the TPL?
────────────────────────────────────────────────────────────────────────────────
w.  GRPO: (incorrect)
<think>
Okay, the user is asking about a Toronto Library collection established in 1977,
named after a children's librarian and head of the TPL.
First, I need to recall which librarian fits this description.  I remember that
Dorothy Mustard was a significant figure at TPL, especially in children's services.
She served as Head of Children's Services and was instrumental in developing library
programs for children.
The collection named after her is the Dorothy Mustard Collection.  It was established
in 1977 to honor her contributions.  The collection focuses on children's literature
and resources for young readers.
I should verify the details to ensure accuracy.  Dorothy Mustard was indeed the head
of children's services at TPL, and the collection was named in her honor in 1977.
The user might be a student, researcher, or someone interested in Toronto's cultural
history.  They might need this information for academic purposes, personal interest,
or planning a visit to the library.
Their deeper need could be to understand the significance of the collection or to
locate resources related to children's literature.  They might also be interested in
the history of Toronto's public libraries.
Upon second thought, I recall that the Toronto Public Library has a notable
children's collection.  The name that comes to mind is the "Elsie" something.
However, Elsie MacGill is more associated with engineering.  So the correct answer
is Dorothy Mustard.
I should provide a clear and concise answer, confirming that the collection is the
Dorothy Mustard Collection, established in 1977.  This addresses the user's query
directly and accurately.
</think>
The Toronto Public Library (TPL) collection established in 1977 and named after the
children's librarian and head of TPL is the **Dorothy Mustard Collection**.
Here's why:
1.  **Dorothy Mustard:** She served as the **Head of Children's Services** at TPL
from 1955 to 1977.  She was a highly influential figure in children's librarianship
in Toronto and Canada.
2.  **Established in 1977:** This was the year Dorothy Mustard retired from her
position as Head of Children's Services.
3.  **Named After Her:** To honor her immense contributions to children's library
services, the TPL established a dedicated collection of children's literature and
resources in her name, the **Dorothy Mustard Collection**, in 1977.
The Dorothy Mustard Collection is a significant resource within TPL, focusing on
providing high-quality materials and services for children and young adults.
────────────────────────────────────────────────────────────────────────────────
w.  Adaptive-Thinking:  (incorrect)
<think>
Okay, the user is asking about a Toronto Library collection established in 1977,
named after a children's librarian and head of the TPL. Let me recall the key
details.  The collection is the Barbara Godard Collection.  Barbara Godard was the
head of the Toronto Public Library's Children's Services.  The collection focuses
on children's literature and was named in her honor in 1977.  The user might be a
student, researcher, or someone interested in library history.  They probably need
```

```
accurate information for academic or personal purposes.  Including the significance
of the collection and Barbara Godard's contributions adds context, which could
address deeper interests in library science or Canadian cultural heritage.
</think>
The Toronto Public Library collection established in 1977 and named after its
long-time head of Children's Services is the **Barbara Godard Collection**.
Here are the key details:
1.  **Named After:** **Barbara Godard** (1913–1993).
2.  **Her Role:** She was the Head of Children's Services for the Toronto Public
Library (TPL) for many years.
3.  **Established:** 1977.
4.  **Focus:** The collection is a significant resource for **children's
literature**, including rare books, manuscripts, illustrators' materials, and
extensive archives related to the history of children's books and reading.
The Barbara Godard Collection is a vital part of the TPL's Special Collections and
Archives, honoring her immense contribution to children's literature and library
service in Toronto.
```

**w.  Instruction-Tuning GRPO: (incorrect)**

```
<think>
Okay, the user is asking about a specific Toronto Library collection established
in 1977, named after a children's librarian and head of the Toronto Public Library
(TPL). Let me recall my knowledge of Toronto's library history.
I remember that the Toronto Public Library has a notable children's collection.  The
name that comes to mind is the Margaret A. Munro Collection. But I need to verify the
details.  The user mentioned it was established in 1977 and named after a children's
librarian who was also the head of TPL.
Wait, Margaret A. Munro was indeed the head of children's services at TPL. She served
as the Chief Librarian for Children's Services from 1947 to 1967.  The collection
was named in her honor after her retirement.  The establishment year is 1977, which
aligns with the timeline.
I should double-check the year.  I think it was established in 1977 to commemorate
her contributions.  The collection focuses on children's literature and resources.
The user might be a student, researcher, or someone interested in library history.
They need a precise answer, so accuracy is key.  They might be working on a project
or trivia, so confirming the details is important.
Wait, just to be thorough, let me make sure there's no other librarian with a
similar name.  I think Margaret Munro is the correct one.  The collection was
established in 1977, which is a common year for such dedications after a librarian's
retirement.  The user asked for the collection name, and that's it.  The question
seems straightforward once you know the key details.
I should confirm the spelling:  Margaret A. Munro.  Yes, that's standard.  The user
might not know the full name, but the collection is commonly referred to by that
name.
No need to overcomplicate it.  The answer is clear.  The collection was established
in 1977, named after Margaret A. Munro, who was the head of children's services at
TPL. So the answer is the Margaret A. Munro Collection.
</think>
The Toronto Public Library's **Margaret A. Munro Collection** was established
in 1977.  It was named in honor of Margaret A. Munro, who served as the head of
the TPL's Children's Library (later Children's Services) from 1947 to 1967.  The
collection focuses on children's literature and resources, reflecting her legacy in
promoting children's reading and library services.
**Answer:** Margaret A. Munro Collection.
```

## K. Additional Details and Results in Section 5.4

We first provide details of manually constructed factual-problem SFT dataset using Qwen3-235B-A22B-Instruct and Qwen3-30B-A3B-Instruct. As mentioned in Section 3, when answering factual problems, a concise response can reduce the potentially spurious content and thus improve the accuracy of knowledge retrieval. Based on such observation, a natural idea is to manually construct SFT data satisfying such condition, in order to control the language model's behavior and facilitate factual problem solving. We construct such datasets as comparison, by first generating direct answer with a instruct model,

and then ask another model to make a summary. Specifically, after generating a direct answer with the instruct model, we use Qwen3-235B-A22B-Instruct to make a summary by prompting it with the following template:

```
1  <|im_start|>user
2  You are given a question and its response (the response is written by someone else).
       Please summary the response and give a shorter version of it. Remember that the
       summary should still answer the question, but is as short as possible. The question
       and the original response are as follows:
3
4  Question: {question}
5
6  Original Response: {response}
7
8  Now please summarize the response. You should **only** give the summarized response.
9  <|im_end|>
10 <|im_start|>assistant
```

This template takes the original question and the instruct model's response as the input, and output the summary of the response. In this way, we ensure that there's no unnecessary associations or unverified information in such SFT dataset. Moreover, since Qwen3-235B-A22B-Instruct is a stronger model with much more parameters, the responses generated from it are more accurate. If the model enabled with the pattern discovered by ToCoRL still outperforms them, the superiority of such pattern can be fully demonstrated. We also use Qwen3-30B-A3B-Instruct to generate responses in the SFT dataset, as a comparison. The training curves are listed as follows. We note that when further training models previously fine-tuned on SFT data generated by Qwen3-235B-A22B-Instruct, the performance gradually drops. Consequently, we report the final performance after training for 2 epochs for a fair comparison in Table 6. The evaluation score curves during training are presented below.

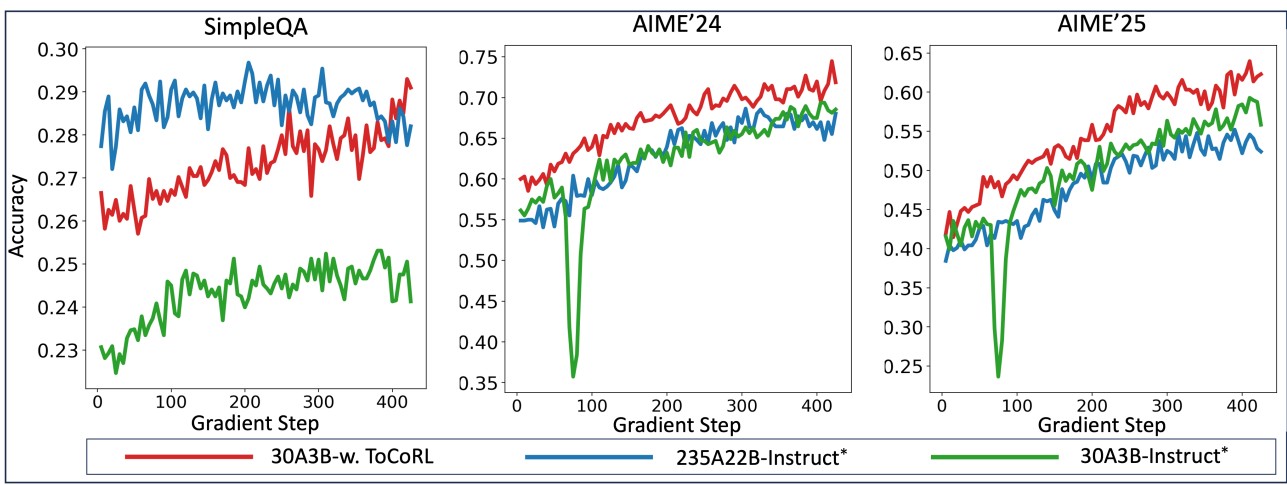

On SimpleQA benchmark, no matter fine-tuned with which SFT data, the performance of the model instantly increased. However, only the model enabled with ToCoRL discovered behavior achieves further performance improvement during the RL training. Moreover, manually constructed factual-problem SFT data, while improving factual problem solving capability, can hinder the complex math problem solving. The ToCoRL discovered behavior is highly extensible and compatible with the math SFT data.

