# OpenReview forum: "Revealing Behavioral Plasticity in Large Language Models: A Token-Conditional Perspective"
_ICML.cc/2026/Conference — ICML 2026 regular_

### Official Review · Reviewer_eCrx · 2026-02-25

**Soundness:** 4
**Presentation:** 3
**Significance:** 4
**Originality:** 4
**Overall Recommendation:** 5
**Confidence:** 4

**Summary:**

This paper studies LLM post-training from a novel and insightful perspective.

Through some empirical analysis, the authors first demonstrate that LLM can switch between different behavioral modes by leveraging its inherent behavioral plasticity and by modifying a prefix of the initial tokens.

To turn this adaption into a stable capability of the LLM, the authors then propose a reinforcement learning method called Token-Conditioned Reinforcement Learning (ToCoRL). Experimental results show that ToCoRL improves accuracy on factuality QA tasks while preserving the reasoning LLM’s ability to solve complex reasoning problems.

This paper also provides analyses of the training dynamics and model behaviors, which are crucial for understanding the underlying behavioral mechanisms.

**Compliance With Llm Reviewing Policy:**

Affirmed.

**Final Justification:**

I keep my original score unchanged as it's already rather high. I think this is a good work.

**Key Questions For Authors:**

- **Extensibility to Multi-Dimensional Behavior Alignment**. Can this approach be extended to simultaneously handling more than two behaviors? The current evaluation effectively demonstrates resolving the conflict between math reasoning and direct factual answering. Could the authors comment or provide preliminary analysis regarding whether ToCoRL can or has the potential to simultaneously handle multiple behavioral modes or tasks? It would be a highly valuable characteristic if ToCoRL can reliably scale to multi-objective alignment.

- **Trade-offs in SFT Data Distillation**. In Section 5.4, the authors prompt the ToCoRL-trained model to generate responses to form a new SFT dataset. Could the authors explain the differences and trade-offs between training on this distilled dataset versus simply combining a factuality QA dataset and a math reasoning QA dataset for SFT?

- **Computational Overhead and LLM-as-a-Judge Costs**. The proposed RL pipeline introduces several resource-intensive components: prefix generation, mixed rollout, and reward computation using LLM-as-a-Judge. Could the authors provide a quantitative breakdown of the computational overhead and financial cost compared to baseline methods?

**Limitations:**

Yes.

**Strengths And Weaknesses:**

## Strengths

- **Novelty and insights**. The method is novel and provides an insightful perspective, suggesting the potential to build a general-purpose language model that can adopt the most suitable behavior for different types of problems.

- **Significant performance improvement and transferability**. Empirically, the method demonstrates effectiveness. It improves the performance of reasoning LLMs on factuality QA task while preserving the ability to handle complex math reasoning. Through training dynamics and case studies, the authors further show how the method mitigates the tendency of reasoning models to produce overly diffuse answers and introduce irrelevant reasoning when addressing factual questions. Moreover, the emergent behavior induced by ToCoRL can be distilled into an SFT dataset and transferred to other models, enabling efficient reuse.

- **Theoretical guarantee**. The paper provides a solid theoretical foundation for ToCoRL.

## Weaknesses

- **Limited evaluation scope**. This work mainly focuses on switching behaviors between factuality QA and mathematical reasoning, but lacking broad evaluation on other task types (e.g., safety, instruction following) and other behavioral dimensions (e.g., style or tone). Although the authors mention computational constraints, demonstrating the framework only on these two tasks is insufficient to prove its robustness as a general-purpose alignment method. I feel even without additional training on other tasks, it would still be valuable to include evaluations on some other typical tasks to ensure that ToCoRL does not induce unforeseen behavioral collapse elsewhere.

- **Insufficient analysis of the underlying mechanisms**. Since the training data come from both factuality QA and math QA, the paper could separately characterize the training dynamics on these two sources and provide clearer comparative analyses, which may show how the emergent behavior on factuality task is internally decoupled from the original reasoning ability.

---

> ### Author Rebuttal · Authors · 2026-03-31
>
> > **W.1:** Limited Evaluation Scope
>
> We provide additional results for Table 2 on other task types. (We do not report Table 3 results because our SFT data does not include these tasks, and the model is not able to handle them.) Specifically, we evaluate performance on IFEval (instruction following), LiveCodeBench-v5 (coding), and WildJailbreak (safety), which are commonly used benchmarks for these domains. The results show that, on tasks without training data, ToCoRL leads to only minimal performance degradation. Moreover, such task-specific performance drops can be mitigated by incorporating corresponding task data during training.
>
> |Table 2|IFEval|LiveCodeBench-v5|WildJailbreak|
> |-|-|-|-|
> |Thinking|87.3|64.5|80.1|
> |Instruct|84.0|41.9|78.7|
> |w. ToCoRL|86.4|63.0|80.6|
> |w. GRPO|87.0|63.8|80.4|
> |w. Adapt-Think|84.7|61.7|79.2|
> |w. I-T GRPO|85.8|62.2|78.5|
>
> > **W.2:** Insufficient Analysis of the Underlying Mechanisms
>
> We further present the training dynamics on math QA (corresponding to Figure 3) and observe several notable findings. We emphasize that math training is consistently conducted using GRPO (as stated in lines 300–305), and the differences arise solely from the influence of factual training under different methods. We provide the figure at the following URL and will include these results and analyses in the revised version.
>
> https://drive.google.com/file/d/1ahUZuF-2V3hFWD1UwixQCSjrZAMKGvMj/view?usp=sharing
>
> For “w. GRPO”, both math and factual tasks are trained using GRPO, so factual training has minimal influence on math performance. As math training progresses, both minimum and average response lengths gradually decrease, while accuracy steadily improves. We believe the reason is that, the initial model is already a strong reasoning model, and further training encourages more efficient reasoning. The training dynamics of “w. ToCoRL” closely mimic those of “w. GRPO” across these metrics, suggesting that ToCoRL introduces no negative impact on math training. In contrast, both “w. Adapt-Think” and “w. I-T GRPO” show negative effects. “w. Adapt-Think” overly reduces response length, harming mathematical reasoning ability. “w. I-T GRPO” exhibits an initial increase followed by a decrease in response length (with accuracy first decreasing and then improving), suggesting that prompt engineering initially biases the model toward increasing reasoning effort for math problems, rather than improving reasoning efficiency (despite the model already possessing strong reasoning capabilities).
>
> > **Q.1:** Extensibility to Multi-Dimensional Behavior Alignment
>
> In ToCoRL, target behavior is injected through the interaction of token-conditional generation and RL. If we aim to inject new behavior for some tasks while preserving original behavior for others, a simple extension would be to apply ToCoRL to selected tasks and standard RL to the rest. Besides, a more stable approach combines per-task ToCoRL training with distilled SFT datasets. For each task, ToCoRL is applied while maintaining standard RL training for all other tasks, until the target behavior stabilizes. Each resulting model is then used to generate a distilled dataset for its corresponding task. Finally, returning to the initial model and performing SFT on the combined distilled datasets, optionally followed by one additional stage of standard RL, yields a model with all desired behaviors.
>
> > **Q.2:** Trade-offs in SFT Data Distillation
>
> The last figure in Appendix K clearly illustrates the difference. Performing SFT on the distilled dataset yields immediate performance gains and strong potential for further scaling via RL training. This demonstrates the value of distilled SFT datasets: they leverage the high transferability of behavioral patterns discovered by ToCoRL, enabling stable, plug-in-style injection of specific behavior patterns without prolonged RL training. The trade-off is that these distilled datasets require a completed ToCoRL training run to obtain.
>
>
> > **Q.3:** Computational Overhead and LLM-as-a-Judge Costs
>
> We would like to clarify that ToCoRL introduces almost no additional computational cost compared to standard GRPO. As shown in Appendix D (Algorithm 1, lines 799–802), the only additional step during training is **pre-generating** a response for each query to serve as the prefix, which is performed prior to RL. This pre-generation is carried out purely in an inference framework, and the resulting responses can be reused across multiple experimental runs, making the additional computation overhead negligible. ToCoRL does not modify model inference, and thus incurs no additional inference cost. The LLM-as-a-Judge setup requires calling the GPT-4o API during training, which introduces some cost. However, this can be replaced with open-source frontier models, since the judge only evaluates the summary portion of LRM outputs. In this case, the additional cost reduces to deploying one extra model for inference, which is manageable.

---

> > ### Author Rebuttal · Reviewer_eCrx · 2026-04-03
> >
> > Thank you for the clarification. Since my initial score is already relatively high, I willkeep it unchanged. Overall, I like this work a lot. Good luck!

---

### Official Review · Reviewer_tH8G · 2026-03-11

**Soundness:** 3
**Presentation:** 3
**Significance:** 3
**Originality:** 4
**Overall Recommendation:** 5
**Confidence:** 4

**Summary:**

The authors tackle the problem for behaviour steering in LRMs; steering the behaviour of LRMs for factual problem solving to be concise by mimicking the behaviour of an instruction finetuned model while at the same time enabling recursive reasoning for mathematical reasoning which benefits from recursive reasoning.

The authors propose Token-Conditioned RL (ToCoRL) which is formulated as a KL regularized online RL algorithm for post-training. In practice the first few tokens from an instruct model are appended to the prompt of the policy before a roll-out and policy gradient step.

The authors test their method using factual QA datasets and mathematical reasoning datasets. The authors compare performance while also measuring the response length and demonstrate improved performance across QA and mathematical reasoning while decreasing response lengths for QA datasets while exhibiting traditional GRPO style reasoning for mathematical reasoning benchmarks.

**Compliance With Llm Reviewing Policy:**

Affirmed.

**Final Justification:**

The authors have addressed certain concerns in my review.

**Key Questions For Authors:**

Does evaluation versus Think or Not? Exploring Thinking Efficiency in Large Reasoning Models via an Information-Theoretic Lens [1] make sense? They also try to reduce the overthinking problem by looking at the entropy of the response? Is there a reason why comparison doesn’t make sense? I think it would be more thorough to compare against this apper and would make the empirical evaluation more thorough.

The broader applicability of ToCoRL is not discussed. I can see how ToCoRL is able to prevent the overthinking issue for LRMs, but this steering framework is more general enabling an LRM to mimic behaviours of any other LLM during post-training. Could it be used for alignment and steering LRMs to be less harmful and more helpful for example. Are there other settings this steering could be useful?

I would be happy to raise my score in light of the rebuttal.

[1] https://arxiv.org/abs/2505.18237

**Limitations:**

The authors have not discussed the limitations of their work. A discussion would strengthen the paper.

**Strengths And Weaknesses:**

# Strengths:
* Really nice idea to frame LRM behaviour steering as KL-regularized RL.
* The ToCoRL objective is well motivated in terms of KL- regularized RL and backed up by theory.
* The results of improved performance on both factual knowledge benchmarks and mathematical reasoning, while reducing answer lengths are pretty convincing (Fig 3).

# Weaknesses:
* In terms of practical implementations, the efficiency tradeoffs versus standard GRPO for instance is not discussed to understand the additional memory and training time requirements of ToCoRL. Since an instruct model is required to condition generation for factual reasoning then more memory is required if both are loaded into memory simultaneously or perhaps steering tokens are generated prior to RL?
* There are certain claims that are not backed up by the experimental results. For instance:
    * Lines 325-327: “Adaptive Thinking and Instruct-Tuning GRPO incur noticeable performance drops on complex math benchmarks”. Personally the drops versus GRPO are modest but more importantly without repeated experiments and standard errors in Table 2, it is difficult to agree with this statement.
    * Lines 425-428: “the LRM equipped with the ToCoRL-discovered behaviour archives the largest accuracy gains and surpasses all baseline models”, this looks the case in table 3 for the mathematical reasoning problems however the gains are less evident for the QA problems. Again, without uncertainties on the results it is difficult to say whether one method is better than another.
* There is a lack of empirical comparison to other work that tries to tackle the overthinking problem in LRMs, for instance [1].

[1] https://arxiv.org/abs/2505.18237

---

> ### Author Rebuttal · Authors · 2026-03-31
>
> > **W.1:** In terms of practical implementations, the efficiency tradeoffs versus standard GRPO ... unless steering tokens are pre-generated prior to RL?
>
> We would like to clarify that ToCoRL introduces almost no additional memory or computational cost compared to standard GRPO. As shown in Appendix D (Algorithm 1, lines 799–802), the only additional step during training is **pre-generating** a response for each query to serve as the prefix, which is indeed performed prior to RL as you suggested. This pre-generation is carried out purely in an inference framework, and the resulting responses can be reused across multiple experimental runs, making both the additional memory cost and training time negligible.
>
> > **W.2:** There are certain claims that are not backed up by the experimental results.
>
> We apologize for not including uncertainty statistics in the original submission due to limited time and computational resources. We have since repeated the experiments in Tables 2 and 6 twice and now report the aggregated results (mean and standard deviation) over three runs as follows:
>
> |Table 2|SimpleQA|Omnisci|AIME’24|AIME’25|
> |-|-|-|-|-|
> |Thinking|18.9|16.1|84.8|80.5|
> |Instruct|22.1|10.5|75.2|61.3|
> |w. ToCoRL|28.4±0.3|18.5±0.2|89.1±0.7|81.3±0.5|
> |w. GRPO|23.8±0.1|16.4±0.3|89.6±0.6|81.4±0.6|
> |w. Adapt-Think|24.0±0.4|16.2±0.3|86.3±0.8|78.6±0.8|
> |w. I-T GRPO|21.3±0.3|14.9±0.4|87.4±0.5|79.8±0.6|
>
> |Table 6|SimpleQA|Omnisci|AIME’24|AIME’25|
> |-|-|-|-|-|
> |30A3B-w. ToCoRL|29.2±0.2|16.0±0.3|74.2±1.2|62.1±0.9|
> |235A22B-Instruct*|28.1±0.2|15.4±0.2|68.2±0.9|55.5±1.1|
> |30A3B-Instruct*|25.5±0.3|14.3±0.4|69.4±1.4|59.0±1.2|
>
> **Regarding Lines 325–327:**
> Since the experiments in Table 2 start from Qwen3-30B-A3B-2507-Thinking, which already achieves strong performance on complex math benchmarks, even "w. GRPO" yields limited absolute improvement. A more meaningful interpretation is to examine each method's relative gain over "w. GRPO" as a baseline. From this perspective, on AIME'24 and AIME'25, "w. ToCoRL" (89.6%, 88.9%) substantially outperforms "w. Adapt-Think" (31.3%, -211%) and "w. I-T GRPO" (54.2%, -77.8%). This conclusion is further supported by the uncertainty statistics reported alongside these results.
>
> **Regarding Lines 425–428:**
> We believe lines 425–428 correspond to Table 6, and we apologize for not reporting the SimpleQA scores before RL training. As shown in the last figure of Appendix K, the ToCoRL-discovered pattern enables a subsequent RL stage to improve accuracy from 26% to 29%, while other methods show marginal or no improvement — demonstrating its greater potential for further RL scaling. The uncertainty statistics above further support this conclusion.
>
> > **W.3 & Q.1:** Additional Baseline Methods: "Adaptive Think"
>
> |Table 1|Accuracy|Avg. Length|Med. Length|
> |-|-|-|-|
> |Normal Gen|18.9|1255|1128|
> |ToCo Gen|20.7|891|477|
> |Adaptive Think Gen ($\alpha=0.1$)|18.3|768|552|
>
> Since “Adaptive Think” is an inference-time method, we compare it with the results in Table 1. The results show that while “Adaptive Think” reduces response length, it also decreases accuracy, which aligns with our findings. As discussed at the end of Section 3, reasoning models often adopt an associative reasoning style, which may introduce spurious content in factual problem solving. Simply truncating reasoning traces based on entropy neither removes already generated spurious content nor changes the underlying behavior that produces it; instead, it disrupts reasoning coherence and degrades performance. In contrast, Figure 3 shows that ToCoRL increases both response length and accuracy in later training stages. Once spurious context is removed via token-conditional generation, effective reasoning patterns naturally emerge through RL optimization. In this sense, ToCoRL more fully leverages the reasoning capabilities of LRMs to improve factual problem-solving. We consider this a meaningful baseline and will include it in the revised version.
>
> > **Q.2:** Broader Applicability
>
> We would like to share a promising application of ToCoRL in steering LRMs toward safer behavior. A safety-aligned LRM, when responding to user queries, may naturally exhibit safety-oriented behaviors at the beginning of its reasoning context (e.g., evaluating the potential harm of the request before deciding how to respond). These behaviors are expressed through generated context and can serve as token prefixes to inject similar behavior into other LRMs. Based on such prefixes, ToCoRL can help internalize this behavior and facilitate the emergence of new, effective, and safe reasoning patterns. We will include a more detailed discussion of the broader applicability of ToCoRL in the updated version.
>
> > **L.1:** Discussion of Limitation
>
> We appreciate the reviewer's suggestion. We have briefly discussed the limitations of this work in Section 7: Conclusion & Discussion, and will expand this into a more thorough discussion in the updated version.

---

> > ### Author Rebuttal · Reviewer_tH8G · 2026-04-03
> >
> > The authors added comparisons to other work I suggested and uncertainties to certain tables.
> >
> > I think this is a good paper.

---

> > > ### Author Response · Authors · 2026-04-03
> > >
> > > Thank you for carefully reviewing our rebuttal. We truly appreciate your constructive feedback and your support in raising the score.

---

### Official Review · Reviewer_tthm · 2026-03-12

**Soundness:** 2
**Presentation:** 2
**Significance:** 2
**Originality:** 2
**Overall Recommendation:** 3
**Confidence:** 3

**Summary:**

This paper proposes ToCoRL, an RL framework that "stabilizes LLM behavioral plasticity" exposed via token-conditional generation. E.g., the model should not generate a long reasoning trace when answering straightforward factual questions. The method uses an instruction finetuned model to generate a few tokens to guide the reasoning model during reinforcement learning, which intuitively leads reasoning models to adapt well to given prompts.

**Compliance With Llm Reviewing Policy:**

Affirmed.

**Final Justification:**

I keep my original score after the rebuttal since my major concern remains unresolved --- the writing of the paper exaggerated its contribution and could be misleading to the readers. The method could have been presented in a much more accurate and concise way.

**Key Questions For Authors:**

- Could you add a simple baseline to compare models with adjustable thinking efforts? E.g., gpt-oss. Otherwise, simply adding a prompt, e.g., "directly give me the answer" at the end of the user prompt, could be a reasonable baseline.
- Since the authors assume the existence of an Instruction-tuned version counterpart of the reasoning model, it would be worth trying to first generate a few tokens with the instruct model, and use these tokens as the prefix for the reasoning model, then continue the generation.

**Limitations:**

yes

**Strengths And Weaknesses:**

Strengths:
The overall idea is simple: using the instruct model-generated answers to steer reasoning traces and stay to the point. And it is shown to improve accuracy on reasoning tasks and reduce token usage.

Weaknesses:
- The writing of the paper is a bit confusing.
I didn't get the chameleon analogy used throughout the first two pages. Is it about adjusting the model behavior in reasoning traces according to the tasks? In my opinion, it would be better to directly get to the technical point of the paper and avoid using analogies, since most readers are very familiar with how language models work.

- Beside, most recent reasoning models support adjusting reasoning efforts now by adding simple flags, which is also a missing baseline of the paper.
- Another missing baseline is to skip RL and force the reasoning model to generate the prefixes during inference. Please see more details below.

---

> ### Author Rebuttal · Authors · 2026-03-31
>
> > **W.1:** The writing of the paper is a bit confusing. I didn't get the chameleon analogy used throughout the first two pages. Is it about adjusting the model behavior in reasoning traces according to the tasks? In my opinion, it would be better to directly get to the technical point of the paper and avoid using analogies, since most readers are very familiar with how language models work.
>
> We apologize for the confusion and provide a clearer explanation of the analogy. A chameleon changes its coloration (behavior) in response to environmental stimuli (external factor). Similarly, the way an LLM addresses a question (behavior) can change under the influence of the token prefix (external factor). We use this analogy to illustrate that a model’s behavior pattern is jointly determined by internal factors (model parameters) and external factors (context prefix). Modifying the context prefix can instantly induce a shift in behavior, but such shifts are transient and unstable. Technically, ToCoRL internalizes and stabilizes these shifts, and more importantly, induces the emergence of new effective behavior patterns. This also applies to non-reasoning models.
>
> ---
>
> ### **Additional Experimental Results**
>
> #### **(W.2, Q.1)**
> GPT-OSS-120B performance on SimpleQA under different reasoning efforts:
>
> |Effort|Accuracy|
> |-|-|
> |high|11.29|
> |medium|10.93|
> |low|10.84|
>
> #### **(W.3, Q.1, Q.2)**
> Comparison of ToCoRL against two additional baselines raised by the reviewer:
> (1) **instruct-tuning**: "simply appending a prompt such as 'directly give me the answer'" ; and
> (2) **token-conditional generation**: "first generating a few tokens using an instruct model, using them as a prefix for a reasoning model, and then continuing generation".
>
> The results are presented as extensions to Tables 2 and 3. For instruct-tuning, we use the appended prompt as described in lines 858–868. For token-conditional generation, since the token prefix is generated by an instruct model, we evaluate on factual benchmarks only.
>
>
> |Table 2|SimpleQA|Omnisci|AIME'24|AIME'25|
> |-|-|-|-|-|
> |w. ToCoRL|28.3|18.6|89.0|81.5|
> |w. I-T GRPO|21.1|14.8|87.8|80.4|
> |instruct-tuning|20.5|13.9|87.6|78.9|
> |token-conditional generation|20.7|15.3|-|-|
>
> |Table 3|SimpleQA|Omnisci|AIME'24|AIME'25|
> |-|-|-|-|-|
> |w. ToCoRL|28.1|15.8|72.1|61.7|
> |w. I-T GRPO|21.7|12.9|56.0|37.8|
> |instruct-tuning|20.1|12.4|56.3|39.0|
> |token-conditional generation|21.3|13.2|-|-|
>
> ---
>
> We believe the reviewer's main concern is the distinction and advantages of ToCoRL over these baselines. Below, we clarify the fundamental differences with experimental evidence.
>
> ### **Analysis of Baselines**
>
> #### **Adjusting reasoning effort (W.2, Q.1)**
> The results above show that lower reasoning effort leads to lower accuracy for GPT-OSS-120B. This supports our key claim: simply reducing reasoning effort does not improve factual problem-solving, as effective reasoning patterns for such tasks are more nuanced. As discussed at the end of Section 5.2, simply shortening the reasoning trace without changing the underlying behavior pattern does not eliminate spurious content (e.g., unnecessary associations), nor does it guarantee improved factual accuracy. Sections 5.2 and 5.3 (especially Figure 3) show that the emergent behavior pattern discovered by ToCoRL combines the elimination of spurious content (initiated by direct answering) with recalibrative reasoning, while ultimately maintaining a relatively high level of reasoning effort.
>
> #### **Instruct-tuning (Q.1)**
> The extended results for Tables 2 and 3 show that instruct-tuning brings only minimal performance gains. In fact, the original results in Tables 2 and 3 already include instruct-tuning combined with RL (I-T GRPO) as a baseline, demonstrating that even integrating instruct-tuning with RL fails to yield meaningful improvements. The underlying reason is that instruct-tuning does not modify the internal factors (model parameters) that govern behavior, and thus can only exert limited influence on behavior patterns.
>
> #### **Token-conditional generation (W.3, Q.2)**
> In fact, the results about this baseline method are also presented in Sections 3. In Section 3, we show that inference-time token-conditional generation can improve performance, but the effects are transient, unstable, and uncontrollable. To remedy this, Section 4 (lines 176–186) develops ToCoRL in a principled way, enabling the emergence of stable and controllable behavior patterns that fully utilize reasoning capabilities.
>
> ### **Summary**
>
> Compared to the baseline methods above, ToCoRL ensures the injection of the target behavior pattern (in factual settings: reasoning without spurious context) while allowing the model to naturally discover the most suitable behavior pattern. The most effective behavior then emerges through the RL process.
>
> If you find our response satisfactory, we would appreciate your consideration in increasing the score.

---

> > ### Author Rebuttal · Reviewer_tthm · 2026-04-04
> >
> > Thanks for the rebuttal! My concerns regarding the missing baselines are resolved.
> >
> > My main remaining concern is regarding the framing of the paper: the first two pages are overreaching and require substantial work to extend the experiments beyond the current 4 benchmarks to support them.

---

> > > ### Author Response · Authors · 2026-04-04
> > >
> > > We respectfully disagree with the characterization that "the first two pages are overreaching." To clarify, we do not claim that ToCoRL achieves equally strong performance across all tasks. We only intuitively motivate the framework as being generalizable — capable of enabling effective behavior steering in alternative settings. We can describe how ToCoRL is instantiated in a concrete alternative task and setting, and provide empirical results as evidence for its generalizability across different settings (which we have done below and will incorporate into the updated version of the paper). However, rigorously instantiating ToCoRL across multiple entirely distinct tasks (each requiring in-depth empirical investigation, analysis, and task-specific algorithmic tuning) demands substantial time and computational resources that scale roughly linearly with the number of tasks. We therefore hope it is understandable that conducting such thorough studies across many tasks is well beyond the page limit of a single paper.
> > >
> > > As a concrete example of ToCoRL's effectiveness on an alternative task, we consider code generation. Language models specialized for coding typically outline their thoughts before producing code — improving accuracy on complex tasks via planning and reasoning — whereas non-specialized models tend to generate code directly. One can extract the initial planning context from specialized models and use it as a prefix to encourage non-specialized models to plan before coding. ToCoRL can then stabilize this behavior, improving performance on complex coding tasks. During the rebuttal period, we conducted experiments in this setting, demonstrating that ToCoRL enables non-specialized models to exhibit plan-before-coding behavior similar to coding-specialized models, and to internalize these shifts, transforming transient inference-time adaptations into stable, learnable behavioral patterns. Full details and results are provided in "Replying to Rebuttal Acknowledgement by Reviewer eMay".
> > >
> > > We hope these results, which extend the experiments in the paper, provide sufficient evidence to address your concern regarding the first two pages. We would be grateful if you would consider revising your score in light of our response.

---

### Official Review · Reviewer_eMay · 2026-03-13

**Soundness:** 2
**Presentation:** 3
**Significance:** 3
**Originality:** 2
**Overall Recommendation:** 4
**Confidence:** 4

**Summary:**

The paper explores behavioral plasticity in Large Language Models (LLMs), showing that they can adapt their behavior at inference time through token-conditional generation. The authors introduce Token-Conditional Reinforcement Learning (ToCoRL), a framework that stabilizes these adaptive behaviors, making them persistent and learnable. The paper demonstrates that ToCoRL enables LLMs to excel in both complex reasoning and factual question answering tasks without retraining. This approach allows for versatile AI systems that can adapt their behavior to different tasks.

**Compliance With Llm Reviewing Policy:**

Affirmed.

**Final Justification:**

The rebuttal addresses most of my concerns.

**Key Questions For Authors:**

1. **Reproducibility:** The paper provides limited information on the training and inference processes. To improve clarity and ensure reproducibility, please provide comprehensive details on these procedures.

2. **Generalization:** Could the authors provide insights into how the framework generalizes across different tasks, model architectures and scales?

3. **Dependence on Prefix Providers:** The reliance on an external "Instruct Model" to provide prefixes introduces a potential point of failure. If the prefix provider is biased or generates hallucinated information, how does this affect the target model's behavior? Exploring how ToCoRL handles noisy or flawed prefix inputs would provide more insights.

**Limitations:**

1. **Scalability and Computational Overhead:** The paper lacks a detailed analysis of the computational cost and scalability of ToCoRL. How does the method impact inference time, memory usage, and other resource constraints in real-world applications? Addressing these concerns would provide a more comprehensive view of the method's practical limitations.

2. **Stability Over Long Contexts:** While ToCoRL aims to stabilize behavioral adaptations, there may be concerns about "behavioral drift" in very long multi-turn conversations, where the initial token prefixes might lose their steering influence. Further discussion on this potential issue would be beneficial.

**Strengths And Weaknesses:**

### Strengths

1. **Clarity and Structure:** The paper is well-organized and clearly written, with a logical progression from the introduction to the experiments. The authors effectively highlight both the theoretical and practical implications of ToCoRL.
2. **Novel Demonstration of Plasticity:** The concept of behavioral plasticity in LLMs is a new and valuable perspective, and the paper shows how this plasticity can be exploited to improve task performance without altering model parameters.
3. **Potential Impact:** The introduction of ToCoRL is a significant contribution to the field of general-purpose AI systems. The framework offers a promising solution to the problem of behavioral adaptability, allowing LLMs to adapt their behavior to different tasks in a seamless and stable manner.

### Weaknesses

1. **Assumptions on Model Behavior and Generalization Across Tasks:** The claim that "Extensive experiments show that ToCoRL enables precise behavioral control without capability degradation" may be overly optimistic. While the authors acknowledge that these behavior shifts depend on external cues (such as token prefixes), this method might not be as effective in tasks where such cues are hard to obtain, such as in creative writing or code generation. Additionally, the mathematical benchmarks used in the paper are limited, as only a few were tested, which doesn't fully demonstrate the robustness of the method.
2. **Narrow Model Testing:** The experiments focus on a single Large Reasoning Model (Qwen3-30B-A3B-2507-Thinking). It would strengthen the paper to test ToCoRL on a broader range of models, including those with different architectures and scales , to better showcase its generalizability.
3. **Deployment Concerns:** The paper doesn't sufficiently discuss practical deployment considerations, such as computational efficiency, scalability, or performance in real-time applications. These practical aspects are crucial for understanding how ToCoRL would perform in real-world environments, especially for large-scale systems.

---

> ### Author Rebuttal · Authors · 2026-03-31
>
> > **W.1:** Assumptions on Model Behavior and Generalization Across Tasks & **Q.2:** Generalization (over tasks)
>
> Although different tasks may require different target behavior patterns, they fundamentally arise from the interaction between model parameters and generated context (as discussed in lines 59–66). When the target behavior is clearly known, it is always expressed through generated context, which can in turn serve as token prefix (external cue) to influence the model's behavior. In this work, we use only a few tokens as the prefix because, for the SimpleQA task, this is sufficient to inject the target behavior (direct answering).
>
> As another example, in code generation, reasoning models typically outline their thoughts before producing code, while non-reasoning models directly output code. One could extract the beginning of such reasoning context and insert it as code comments, using them as a prefix to encourage non-reasoning models to plan before coding. ToCoRL can then help stabilize this behavior, improving performance on complex coding tasks. While we are unable to provide numerical results due to pipeline complexity and computational cost, we hope this example illustrates the generality of ToCoRL.
>
> The additional mathematical benchmark results in Tables 2 and 3 show that ToCoRL has minimal impact on math problem-solving capability.
>
> |Table 2|AMC'23|MATH-500|
> |-|-|-|
> |Thinking|99.4|98.8|
> |Instruct|95.6|97.8|
> |w. ToCoRL|99.4|98.6|
> |w. GRPO|99.5|98.5|
> |w. Adapt-Think|98.4|98.0|
> |w. I-T GRPO|98.6|98.1|
>
> |Table 3|AMC'23|MATH-500|
> |-|-|-|
> |Base|60.7|65.1|
> |SFT|91.3|95.6|
> |w. ToCoRL|99.0|98.4|
> |w. GRPO|99.1|98.7|
> |w. Adapt-Think|96.2|96.3|
> |w. I-T GRPO|97.5|97.0|
>
> > **W.2:** Narrow Model Testing and Generalization on Model Scale & **Q.2:** Generalization (over model architecture and scale)
>
> We repeat the experiments in Tables 1 and 2 using DeepSeek-R1-Distill-Llama-70B. The results still show consistent superiority of ToCoRL.
>
> |Table 1|Accuracy|Avg. Length|Med. Length|
> |-|-|-|-|
> |Normal Gen|16.8|874|678|
> |ToCo Gen|17.9|135|91|
>
> |Table 2|SimpleQA|Omnisci|AIME’24|AIME’25|AMC'23|MATH-500|
> |-|-|-|-|-|-|-|
> |DeepSeek-R1-Distill-Llama-70B|16.8|17.6|68.3|51.5|94.5|93.8|
> |w. ToCoRL|26.5|18.9|80.1|63.8|96.2|95.4|
> |w. GRPO|20.6|17.9|80.3|64.5|96.8|95.2|
> |w. Adapt-Think|21.2|17.2|72.5|55.7|94.9|94.7|
> |w. I-T GRPO|19.8|16.0|75.2|57.3|95.2|94.3|
>
> > **W.3:** Deployment Concerns & **L.1:** Scalability and Computational Overhead
>
> ToCoRL introduces no additional inference overhead, as token-conditional generation is used only during training. Once the target behavior is internalized by the model through training, it is expressed naturally at inference. The only extra training cost is pre-generating a prefix per query, which can be reused across runs and is negligible. As RL-based training is already standard in LLM development, ToCoRL is well-suited for practical deployment.
>
> > **Q.1:** Reproducibility
>
> We provide additional training and inference configurations:
> Training:
> - Codebase: modified verl
> - Reward compute: Qwen2.5-Math
> - Prefix pre-generation: Engine: vllm, top_p=0.8, top_k=20, temperature=0.7
> - RL rollout: Engine: vllm, top_p=1.0, top_k=-1, temperature=1.0, max generation length=28762
> - Update engine: megatron
>
> Inference:
> - Accuracy calculation: Qwen2.5-Math
> - Engine: vllm, top_p=0.95, top_k=20, temperature=0.6
>
> > **Q.3:** Dependence on Prefix Providers
>
> A key strength of ToCoRL is its robustness to noisy or flawed prefixes, enabled by correctness-aware guided exploration and reinforcement learning, both driven by correctness. In correctness-aware guided exploration, positive samples exhibiting the target behavior are reinforced, while negative samples containing harmful context (e.g., induced by noisy or flawed prefixes) are suppressed. Reinforcement learning further eliminates the impact of suboptimal prefixes through reward maximization. This is supported by Table 5: although Qwen3-30A3B-2507-Instruct (SimpleQA Accuracy: 22.1%) is much weaker than Qwen3-235BA22B-2507-Instruct (SimpleQA Accuracy: 52.9%) on factual questions, and its prefixes may include spurious context, the final training performance remains largely unaffected.
>
> > **L.2:** Stability Over Long Contexts
>
> Ideally, even in multi-turn settings, the target behavior should still be reflected in the context, though token-prefix selection could be more nuanced. A straightforward approach is to extend the prefix length, which is expected to better utilize ToCoRL in multi-turn scenarios. A more promising alternative is to have the prefix provider iteratively generate subsequent prefixes across multiple turns based on the dialogue history. While potentially more effective, this approach incurs higher computational cost. We believe extending ToCoRL to such scenarios is both meaningful and necessary, and we leave it as future work
>
> If you find our response satisfactory,we would appreciate your consideration in increasing the score

---

> > ### Author Rebuttal · Reviewer_eMay · 2026-04-04
> >
> > Thank you for the rebuttal. It addresses several of my original concerns. In particular, the additional experiments on DeepSeek-R1-Distill-Llama-70B strengthen the empirical support for the method. My main remaining concern is the gap between the breadth of the paper's claims and the scope of its empirical validation. While the paper positions ToCoRL as a broadly applicable framework for behavior adaptation, the experimental evidence is still concentrated on the specific setting of factual QA with math reasoning.

---

> > > ### Author Response · Authors · 2026-04-04
> > >
> > > We apologize for not including empirical results on the code generation task in our initial rebuttal due to time and budget constraints. We have since conducted relevant experiments, and the results are presented below. Remarkably, ToCoRL also effectively leverages LLM behavior plasticity on this coding task, internalizing the plan-before-coding behavior of coding-specialized language models into non-specialized models as stable and learnable behaviors.
> > >
> > > Specifically, we experiment with Qwen2.5-Coder-32B-Instruct and Qwen2-7B-Instruct. Qwen2.5-Coder-32B-Instruct explicitly reasons before coding (first analyzes the problem and formulates a plan, then produces code). In contrast, Qwen2-7B-Instruct directly outputs code. We illustrate this behavioral difference with a concrete example:
> > >
> > > **Problem:**
> > > ```
> > > Write a Python function `parse_nested_parens(paren_string: str) -> List[int]` to solve the following problem:
> > > Input to this function is a string represented multiple groups for nested parentheses separated by spaces.
> > > ...
> > > ```
> > > **Qwen2.5-Coder-32B-Instruct** (reasoning+code):
> > > ```
> > > To solve the problem of finding the deepest level of nesting of parentheses for each group in a given string, we can follow these steps:
> > >
> > > 1. Split the input string by spaces to get individual groups of parentheses.
> > > 2. For each group, iterate through the characters and keep track of the current level of nesting using a counter.
> > > 3. Update the maximum level of nesting encountered for each group.
> > > 4. Store the maximum level of nesting for each group in a list.
> > > 5. Return the list of maximum nesting levels.
> > >
> > > Here's the implementation of the `parse_nested_parens` function:
> > >
> > > ```python
> > > from typing import List
> > >
> > > def parse_nested_parens(paren_string: str) -> List[int]:
> > >     ...
> > > ```
> > > **Qwen2-7B-Instruct** (direct code):
> > > ```
> > > ```python
> > > from typing import List
> > >
> > > def parse_nested_parens(paren_string: str) -> List[int]:
> > >     ...
> > > ```
> > > To induce behavioral pattern shift through token-conditional generation, we adopt the **"First k Sentences"** strategy: we split by `\n` and use the first k sentences from Qwen2.5-Coder-32B-Instruct as a token prefix to steer Qwen2-7B-Instruct. We first conduct inference-only experiments analogous to Table 1, evaluating on HumanEval and LiveCodeBench (v1):
> > > |Qwen2-7B-Instruct|HumanEval|LiveCodeBench|
> > > |-|-|-|
> > > |Normal Gen|78.7|23.9|
> > > |First 1 Sentence|81.3|25.1|
> > > |First 2 Sentences|82.8|25.9|
> > > |First 3 Sentences|83.2|25.6|
> > > |First 4 Sentences|83.1|25.8|
> > >
> > > The results show that at inference time, token-conditional generation yields immediate performance gains. The prefix encourages Qwen2-7B-Instruct to plan in natural language before coding, improving its problem-solving capability (aligned with our intuition). Performance gains saturate at around k=3. Notably, the first sentence from Qwen2.5-Coder-32B-Instruct consistently takes the form *"To solve the problem of ..., we can follow these steps:"* — introducing no concrete planning content — yet Qwen2-7B-Instruct still demonstrates its own planning capability and achieves improved performance, which is highly consistent with our findings. A concrete example is provided below:
> > >
> > > **"First 1 Sentences" Strategy:**
> > > ```
> > >  To solve the problem of finding the deepest level of nesting of parentheses for each group in a given string, we can follow these steps:
> > >
> > > 1. Split the input string into groups based on spaces.
> > > 2. For each group, initialize a variable `depth` to 0 to keep track of the current nesting level.
> > > 3. Iterate through each character in the group:
> > >    - If the character is an opening parenthesis `(`, increment `depth` by 1.
> > >    - If the character is a closing parenthesis `)`, decrement `depth` by 1.
> > >    - Keep track of the maximum `depth` encountered during the iteration for each group.
> > > 4. Append the maximum `depth` for each group to a list.
> > > 5. Return the list of maximum depths.
> > >
> > > Here's the implementation of the `parse_nested_parens` function:
> > >
> > > ```python
> > > from typing import List
> > >
> > > def parse_nested_parens(paren_string: str) -> List[int]:
> > >     ...
> > > ```
> > > Moreover, we also conducted full ToCoRL training under this setting. Due to limited computational budget, we trained only on coding problem queries to demonstrate that ToCoRL remains effective in this entirely different setting. We construct 40K training queries from open-source datasets (e.g., rStar-Coder, TACO, etc.). Based on them, we compare ToCoRL against GRPO. We note that we use the above “First 1 Sentence” strategy for token-conditional generation, and the evaluation benchmarks remain HumanEval and LiveCodeBench. The results are as follows:
> > > ||HumanEval|LiveCodeBench|
> > > |-|-|-|
> > > |Qwen2-7B-Instruct|78.7|23.9|
> > > |ToCoRL|90.9|31.2|
> > > |GRPO|84.1|26.0|
> > >
> > > The results clearly demonstrate the advantage and greater potential of the "planning before coding" behavioral pattern enabled by ToCoRL, further validating its generalizability across diverse settings.
> > >
> > > If you find our response satisfactory, we would appreciate your consideration in increasing the score.

---

### Decision · Program_Chairs · 2026-04-30

**Decision:**

Accept (regular)

**Comment:**

This paper introduces Token-Conditioned Reinforcement Learning (ToCoRL) to control the behavior of large reasoning models. By conditioning RL on short token prefixes, it teaches models to dynamically switch between deep reasoning (for complex math) and concise answering (for factual QA), effectively solving the "overthinking" problem without adding inference overhead.

Strengths:

Novel Framework: Elegant, mathematically sound use of token-conditional generation to guide KL-regularized RL.

Strong Results: Successfully resolves the tension between factual QA accuracy and advanced mathematical reasoning (e.g., AIME, MATH-500).

Outstanding Rebuttal: The authors provided extensive new experiments, proving generalization to larger scales (DeepSeek-R1-70B), new domains (code generation, safety), and addressing all requested baselines.

Weaknesses:

Pipeline Dependency: Training requires an external instruction-tuned model to pre-generate the steering prefixes.

Initial Framing: The initial "chameleon" analogy and broad claims slightly overreached the original experiments, though the rebuttal heavily mitigated this concern.

This paper directly addresses a timely challenge in LLM post-training (reasoning models overthinking simple queries). The method is effective, incurs zero extra inference cost, and the authors' exceptionally thorough rebuttal makes it a clear accept for ICML.